# Deconvolution improves the detection and quantification of spike transmission gain from spike trains

Lidor Spivak[1], Amir Levi [1], Hadas E. Sloin [1], Shirly Someck[1] & Eran Stark [1✉]

Accurate detection and quantification of spike transmission between neurons is essential for determining neural network mechanisms that govern cognitive functions. Using point process and conductance-based simulations, we found that existing methods for determining neuronal connectivity from spike times are highly affected by burst spiking activity, resulting in over- or underestimation of spike transmission. To improve performance, we developed a mathematical framework for decomposing the cross-correlation between two spike trains. We then devised a deconvolution-based algorithm for removing effects of second-order spike train statistics. Deconvolution removed the effect of burst spiking, improving the estimation of neuronal connectivity yielded by state-of-the-art methods. Application of deconvolution to neuronal data recorded from hippocampal region CA1 of freely-moving mice produced higher estimates of spike transmission, in particular when spike trains exhibited bursts. Deconvolution facilitates the precise construction of complex connectivity maps, opening the door to enhanced understanding of the neural mechanisms underlying brain function.

[1] Sagol School of Neuroscience and Department of Physiology and Pharmacology, Sackler Faculty of Medicine, Tel Aviv University, Tel Aviv 6997801, Israel.
✉email: eranstark@tauex.tau.ac.il

Neurons in the brain act as information gates, each receiving inputs from thousands of different sources and deciding which to pass on to other neurons. The capability of transmitting relevant inputs to other cells is at the very core of sensory processing, action, and cognition. One way for investigating how information is transmitted between two neurons is to measure the postsynaptic potential (PSP) in one cell following a spike of a presynaptic cell[1–3]. When excitatory PSP (EPSP) magnitude is larger, the probability that the postsynaptic neuron will fire following a presynaptic spike is higher[2]. However, EPSP magnitude is affected not only by synaptic strength, but also by the instantaneous membrane potential of the postsynaptic neuron[4]. Moreover, even constant EPSP magnitude cannot predict whether a postsynaptic neuron will fire after a presynaptic spike. When an EPSP is generated at lower membrane potentials, cells are less likely to fire[5]. Thus, EPSP magnitude alone is insufficient for determining whether a postsynaptic neuron will fire following a presynaptic spike, thereby allowing information to pass on to other cells.

An alternative approach for determining connectivity between two neurons is to directly quantify whether presynaptic spikes generate spikes in the postsynaptic neuron. The spike-to-spike cross-correlation histogram (CCH)[6–8] has been used for measuring the transmission of spikes between two neurons[9–11]. However, most CCH applications employ spontaneous spiking, and are therefore correlative measurements which cannot differentiate causal transmission from non-causal spike patterns. For example, firing of two cells may be co-modulated by another source such as visual input or a third neuron, causing the two cells to fire in synchrony[12]. Thus, the CCH is influenced not only by spike transmission properties but also by co-modulation of the pre- and postsynaptic trains[13–15].

To differentiate between direct spike transmission due to inter-neuronal coupling and other indirect sources, several methods have been proposed[16–20]. An underlying assumption in these approaches is that while direct connections exhibit fast transients in the CCH, indirect sources produce slower fluctuations. Based on this assumption, timescale separation can be used to detect and estimate synaptic connectivity, differentiating between fast transients (which are presumably due to direct connectivity), and slower fluctuations (which are presumably due to other sources). Indeed, when timescale separation methods are employed, spike transmission measurements derived from casual methods are consistent with measures derived from spontaneous spiking CCHs[21]. However, it is still unclear to what extent timescale separation methods can differentiate between transmitted spikes and third-party, synchronous spiking. Moreover, the CCH is influenced not only by the interaction between the pre- and postsynaptic spike trains, but also by the spiking activity pattern of each neuron[8]. When one of two connected neurons exhibits high-frequency periodicity or burst spiking, fast features appear in the CCH[22,23]. These features cannot be differentiated from transmitted spikes by timescale separation. Thus, even when timescale separation methods are used, transmitted spikes cannot be fully differentiated from non-transmission patterns such as burst spiking activity.

To remove non-transmission features from the CCH and improve the estimation of functional connectivity from spike trains, we formulated a mathematical framework for CCH composition. We found that the CCH can be expressed as a sum of three distinct elements. Based on this finding, we developed a deconvolution-based algorithm for removing second-order spike train statistics from the CCH. Using deconvolution in concert with timescale separation methods removed the effect of burst spiking, improving the accuracy of both detection and quantification of neuronal connectivity in synthetic data. Consistent with simulation results, application to neuronal data recorded from CA1 of freely-moving mice showed that deconvolution increased estimates of spike transmission, in particular when the spike trains exhibited bursts.

## Results

**Spike transmission gain between two neurons can be accurately estimated from spike-to-spike cross-correlation.** Neurons can transmit spiking information to other neurons via different connections including chemical synapses (e.g., AMPA, GABA, NMDA, or other receptors)[24,25], electrical synapses (gap junctions)[26], or ephaptic coupling (electromagnetic fields)[27]. Whether spiking information will propagate in the brain depends on the effective connectivity between neurons. When membrane potentials are accessible, connectivity between two neurons can be quantified by the magnitude of the EPSP in the postsynaptic neuron following a single presynaptic spike[1–3]. However, EPSP magnitude does not indicate whether the postsynaptic neuron will pass information onward to other neurons in the form of spikes. Alternatively, the spike trains themselves may be used for estimating the effective connectivity between two neurons, defined here as spike transmission.

To determine the relations between EPSP-based estimates and spike transmission, we simulated a two-cell network in which coupling strength was varied (Fig. 1). The presynaptic neuron exhibited Poisson spiking modified by a 2 ms refractory period ($\lambda_1 = 5$ spk/s). The postsynaptic neuron was modeled as a conductance-based leaky integrate and fire (LIF) neuron. In the lack of connectivity, the postsynaptic neuron did not emit any spikes when membrane potential variability, quantified using $\sigma$, was low (Fig. 1a). When $\sigma$ was increased, spontaneous spike rate increased monotonically (Fig. 1a). Specifically, in the range of noise observed in intact preparations ($\sigma < 4$ mV)[5,28], spike rate was in the range observed in freely-moving animals (<25 spk/s)[29].

Next, we connected the pre- and postsynaptic neurons via a synaptic conductance. Conductance magnitude was calibrated to produce unitary EPSPs (uEPSPs) in the range of 0–2 mV, as observed in vivo[2,28,30,31]. When membrane potential variability was low ($\sigma < 2$ mV), even the stronger uEPSPs failed to generate postsynaptic spiking. However, when $\sigma$ was higher (2–4 mV), uEPSPs of the same magnitude induced postsynaptic spikes in addition to the spontaneous noise-induced spikes (Fig. 1b, red curve). To quantify effective connectivity, we defined "real spike transmission gain" (rSTG) as the number of extra postsynaptic spikes generated following every presynaptic spike. In the range of biologically-relevant noise, rSTG increased monotonically with $\sigma$. Furthermore, for every $\sigma$, rSTG increased monotonically with uEPSP magnitude (Fig. 1b, top inset). While even relatively-large EPSPs fail to produce postsynaptic spikes in the lack of membrane potential variability, weaker EPSPs (0.2–1 mV) could induce consistent spiking when noise was sufficiently high. Moreover, the uEPSP magnitude required to generate a fixed rSTG decreased monotonically with $\sigma$ (Fig. 1b, bottom inset). Thus, EPSP magnitude alone does not suffice for estimating spike transmission.

To determine whether effective connectivity associated with weak EPSPs can be detected from spike timing alone, we constructed CCHs for pre- and postsynaptic spike train pairs (Fig. 1c). We defined the "spike transmission curve" (STC) as the impulse response of spike transmission between the pre- and postsynaptic neurons. Empirically, an estimated STC (eSTC) can be determined by the difference between the CCH and the baseline. Here, we defined the baseline using the "tails" predictor, namely the CCH count at longer time lags ($|\tau| \geq 11$ ms). We then defined the estimated STG (eSTG) as the area under the STC peak

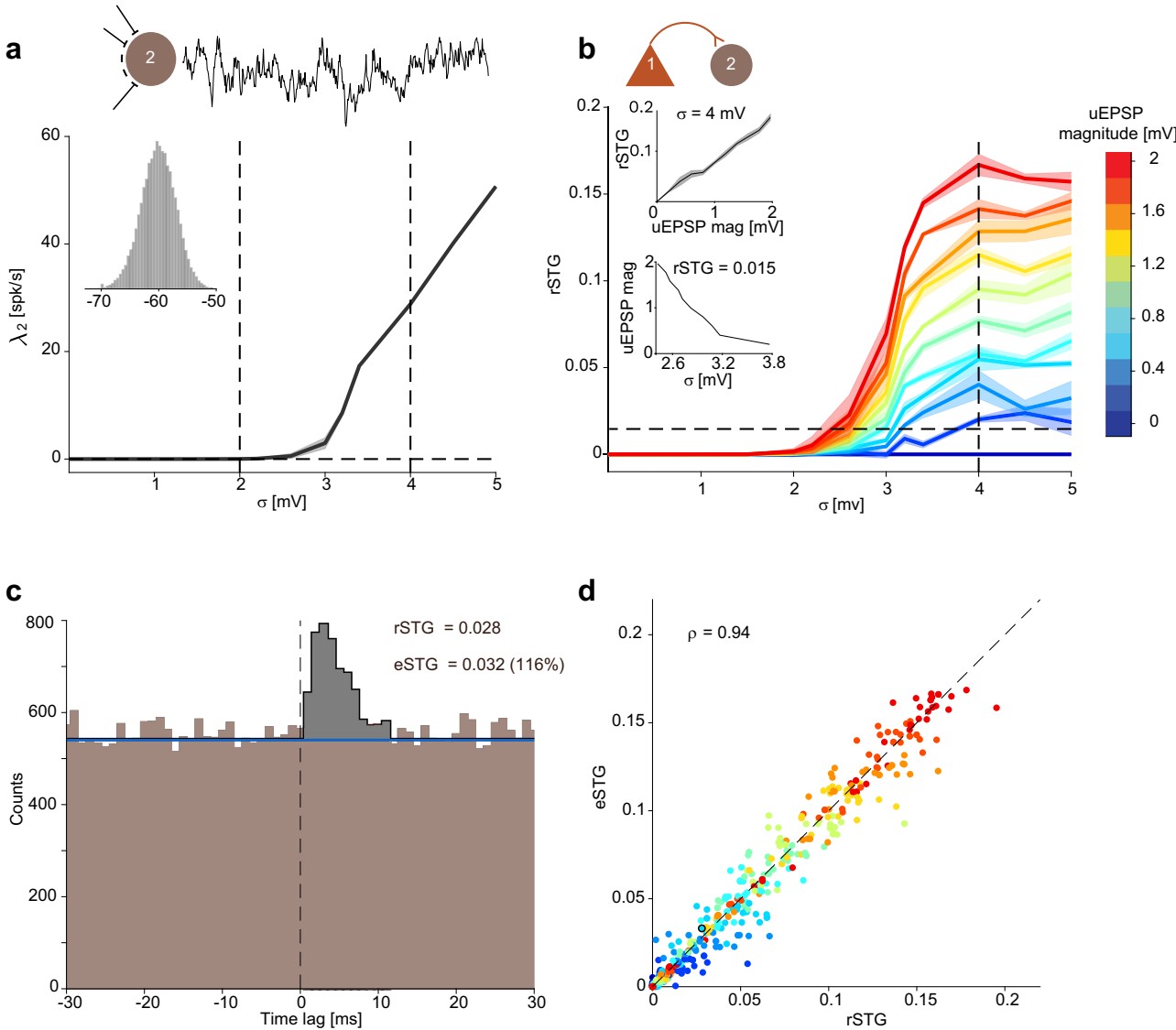

**Fig. 1 Spike transmission between two neurons can be estimated from spike times. a** Increased membrane potential variability increases spontaneous spiking. A leaky integrate and fire (LIF) model neuron was simulated with different levels of membrane potential variability, quantified by the noise SD, $\sigma$ ($n = 6$ repetitions for each of $n = 11$ $\sigma$ values, from 0 to 5 mV with 0.5 mV increments; each 180 min). The firing rate ($\lambda_2$) of the simulated neuron is shown for each $\sigma$ value; here and in (**b**), error bands indicate SEM. The trace next to the cartoon shows 300 ms of simulated membrane potential, and the histograms show the distribution of membrane potentials ($\sigma = 3$ mV). **b** Even small-magnitude EPSPs induce observable spike transmission gain in the presence of noise. The simulated LIF neuron (2) was driven via an excitatory synapse by a presynaptic neuron (1) with Poisson spiking ($\lambda_1 = 5$ spk/s) modified by refractoriness. Synaptic conductance was modified to yield $n = 11$ different unitary EPSP (uEPSP) magnitudes (from 0 to 2 mV with 0.2 mV increments), and the real spike transmission gain (rSTG) was computed for every $\sigma$ and uEPSP combination ($n = 121$). Both uEPSP magnitude and membrane potential variability control spike transmission. **c** A cross-correlation histogram (CCH; bins size: 1 ms) was computed from a pair of simulated spike trains as in (**b**); $\sigma = 3.4$ mV, uEPSP magnitude = 0.6 mV. For deriving an estimated STG (eSTG) from the CCH, the level of baseline joint counts (blue line) was first estimated using the "tails" predictor, averaging CCH values at time lags $|\tau| \geq 11$ ms. The spike transmission curve (STC; black line) was estimated as the above-baseline curve with a peak in the temporal region of interest (ROI; defined here as $0 < \tau \leq 5$ ms; gray bins), with zeros in all other bins. The eSTG is the area under the estimated STC. **d** Spike transmission gain estimated from the CCH is an accurate estimate of the real STG. For every simulation run ($n = 6$ repetitions for each of 121 $\sigma$ and uEPSP combinations), rSTGs and eSTGs were computed. Different colors correspond to different uEPSP magnitudes as in (**b**), and the black circle corresponds to the example in (**c**). Spearman's correlation coefficient between eSTG and rSTG is $\rho = 0.94$ ($n = 726$ random samples, $p < 0.001$, permutation test).

in a temporal region of interest (ROI; defined here as $0 < \tau \leq 5$ ms[32]; Fig. 1c, gray bins), divided by the number of presynaptic spikes. For an STC below baseline, the STG is a negative quantity. Previous studies have referred to the STG using various other names: "effectiveness"[33], "asynchronous gain"[34], "synaptic efficacy"[11], and "spike transmission probability"[21,32]. For the CCH in Fig. 1c, generated using relatively weak EPSPs

(0.6 mV; $\sigma = 3.4$ mV), the eSTG (0.032) was of the same order of magnitude as the rSTG (0.028, 116%; Fig. 1c). Considering all non-zero rSTG values, the mean eSTG error (eSTG–rSTG)/rSTG, was 2.5% (456 spike train pairs with non-zero rSTG; smaller than 5%, $p < 0.05$, one-tailed Wilcoxon's signed-rank test; Fig. 1d). Thus, the STG estimated from the CCH between spike trains closely matches the real spike transmission gain.

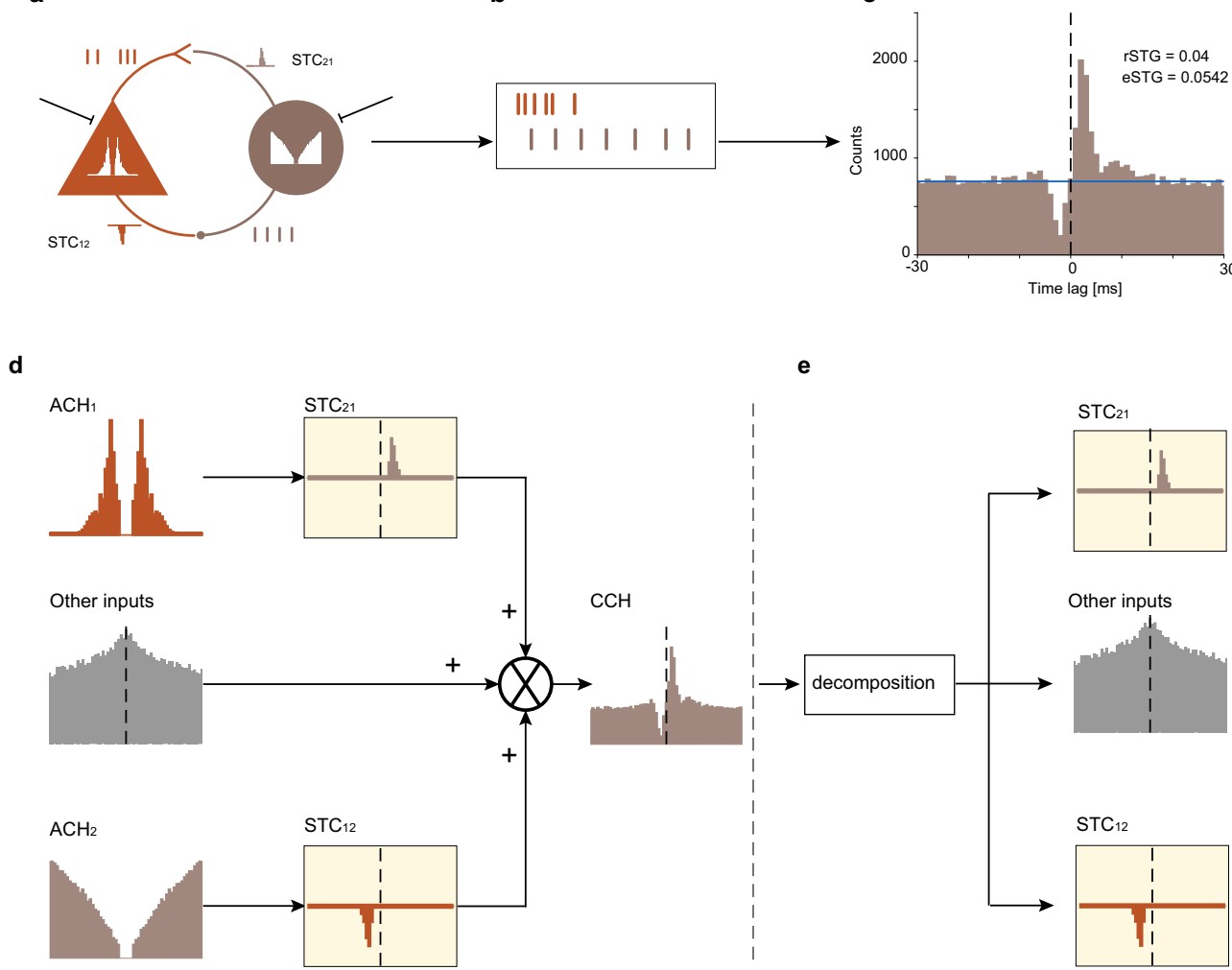

**Fig. 2 The cross-correlation between two spike trains can be expressed as a sum of three elements. a** Cartoon of two neurons, driven by external sources and coupled by reciprocal excitatory and inhibitory monosynaptic connections. In the example, both neurons exhibit spiking activity which deviates from a Poisson process: the excitatory neuron exhibits burst spiking, and the inhibitory neuron exhibits second-order gamma spiking activity. **b** The spike trains of the two neurons result from all of these (and possibly other) sources. **c** An example CCH created from the synthetic spike trains described in (**a**, **b**). The rSTG is 0.04, whereas the eSTG provides an inaccurate estimate (0.054; 135%). The example demonstrates that in the presence of complex dynamics, the STG cannot be estimated accurately by the "tails" method. **d** The CCH between two coupled spike trains, $s_1$ and $s_2$, can be expressed as a sum of three elements. The first element is the convolution of the auto-correlation histogram of the first train (ACH$_1$) with the STC from neuron 1 to neuron 2 (STC$_{21}$). The second element, denoted as "other inputs", is equal to the cross-correlation between the uncoupled spike trains. The third element is the convolution of ACH$_2$ with STC$_{12}$. **e** Based on the composition (**d**), a decomposition procedure may recover the other inputs and the two STCs from the CCH.

**The cross-correlation between two spike trains can be expressed as a sum of three elements**. The foregoing discussion (and Fig. 1) focused on a specific configuration in which the presynaptic spike train was generated via a homogenous Poisson process modified by refractoriness. Furthermore, all statistical dependencies between the two spike trains were captured by a single modeled synapse. In general, dependencies within and between spike trains may be richer (Fig. 2a). Individual spike trains typically deviate from Poisson processes, conforming to higher-order gamma processes or exhibiting spike bursts[35,36]. Connectivity between neurons is often not purely feedforward, exhibiting reciprocal excitation[1] or feedback inhibition[37,38]. Furthermore, dependencies between two spike trains may result from covariance of other inputs to the two neurons[6,8,14,39] rather than from synaptic connections between the neurons. These and other possible sources of complexity give rise to the observed spike trains (Fig. 2b) from which the CCH is derived (Fig. 2c).

Therefore, the STC estimated from a pair of spike trains is generally not identical to the real STC. As a direct consequence, the eSTG is not identical to the rSTG.

In general, the STC is not necessarily identical to the difference between the CCH and the baseline. However, if the STC is fixed (more precisely, if the system is linear time-invariant), there is an exact mathematical dependency between the STC and the CCH. In the absence of external input to the postsynaptic neuron, all postsynaptic spikes ($s_2$) are due to spontaneous presynaptic spikes ($s_1^0$). In that case, the CCH is identical to the convolution (*) between the spike-to-spike auto-correlation histogram (ACH) of neuron 1 spikes, and the STC from neuron 1 to neuron 2 (Fig. 2d):

$$CCH = ACH_1 * STC_{21} \qquad (1)$$

When external input is added to the postsynaptic neuron ($s_2^0$), the CCH is equal to the sum of the above convolution and the

cross-correlation ($\star$) between the spike trains which are not due to the direct connectivity:

$$CCH = ACH_1 * STC_{21} + s_1^0 \star s_2^0 \qquad (2)$$

When the connectivity between the neurons is bidirectional, and external (possibly correlated) inputs are added to both neurons, the CCH can be approximated as a sum of three elements:

$$CCH = ACH_1 * STC_{21} + s_1^0 \star s_2^0 + ACH_2 * STC_{12} \qquad (3)$$

The approximation is valid as long as the product of the two STGs (i.e., the loop gain) is small. Thus, a CCH can be described as a sum of three elements: (1) the convolution between the ACH of neuron 1 and the impulse response from neuron 1 to neuron 2; (2) the convolution between the ACH of neuron 2 and the impulse response from neuron 2 to neuron 1; and (3) the cross-correlation between the background spike trains, those not due to direct connectivity between the two neurons. The last expression has the conceptual benefit of opening the loop, effectively describing a feedback system (Fig. 2a) as a feedforward system using an arithmetic decomposition (Fig. 2e). Moreover, understanding that the CCH is a sum of three elements allows reconstructing hidden elements, namely the STCs, from the CCH using numerical methods.

**Deconvolution eliminates burst spiking effects and enables accurate STG and PSP estimation.** To estimate inter-neuronal effective connectivity from spike trains, several methods have been proposed[20,32,40–42]. These and other procedures rely on the concept of timescale separation, in which processes that occur at fast timescales (e.g., monosynaptic spike transmission) are distinguished from processes occurring at slower timescales (e.g., co-modulated other inputs, such as external inputs or network oscillations). However, in addition to other inputs, the CCH is comprised of the convolution products of each ACH with the corresponding STC (Fig. 2d), which occur at fast timescales. Hence, even if an optimal estimate of the other inputs is removed, a CCH between two synaptically-connected neurons will still contain contributions from the ACHs of each neuron.

To demonstrate the effect of second-order spike train statistics on connectivity estimation, we simulated two spike trains coupled by an excitatory monosynaptic connection (rSTG = 0.04; 833 min; Fig. 3a–c). The spike train of the presynaptic neuron was initially simulated as a Poisson process modified by an absolute refractory period (ARP; $ARP_1 = 2$ ms; $\lambda_1 = 2$ spk/s), and exhibited bursting activity (Fig. 3b). The extent of bursting was controlled by a "burst fraction" (BR) parameter, defined as the probability of each spike to be followed by another spike within a short inter-spike interval (3–7 ms). BR was set to 0.4. The spike train of the postsynaptic neuron realized a second-order gamma process modified by refractoriness ($\lambda_2 = 8$ spk/s; $ARP_2 = 2$ ms; Fig. 3a). Consistent with the construction, the CCH between the coupled spike trains exhibited a peak within the causal ROI (Fig. 3c). In addition, the CCH exhibited side lobes: secondary peaks with increased counts on both sides of the causal peak, which are due to the bursting activity of the presynaptic neuron. Similar patterns were previously reported in real datasets[22].

To quantify the effect of the ACH on the estimation of effective connectivity, we used five different methods: tails, jitter, median, GLMCC[40], and CoNNECT[41]. While the tails, jitter, and median filtering methods yield estimates of effective connectivity measured as gain (STG; Fig. 1), the GLMCC and CoNNECT methods produce estimates in units of mV (PSP). The estimates yielded by the PSP-based methods cannot be gauged directly in the case of the synthetic CCH depicted in Fig. 3c, since no ground

truth for PSP exists in the point process simulations. The tails predictor overestimated the rSTG (eSTG = 0.053, 133%; Fig. 3d), and the jitter and median predictors underestimated the rSTG (jitter: 0.023, 58%; median filtering: 0.028, 70%). Thus, when burst spiking activity is present, the tails, jitter, and median methods fail to produce an accurate eSTG.

For accurate estimation of effective connectivity from the CCH in the presence of arbitrary second-order spike train statistics (e.g., bursting or periodicity), we developed a deconvolution algorithm that takes advantage of the CCH mathematical properties described in Fig. 2d, e. In a nutshell, the algorithm mitigates the effect of bursting activity by deconvolving the ACHs from the CCH, resulting in a CCH which does not exhibit ACH-induced side lobes (Fig. 3f). The outcome is a "deconvolved CCH" (dcCCH). The dcCCH still includes the additive effect of other inputs (Fig. 2d). However, the other inputs element can be estimated using timescale separation methods and removed by subtraction. Thus, deconvolving the ACHs from the CCH before applying timescale separation removes ACH artifacts, and may improve connectivity estimates.

To test whether deconvolution indeed improves STG/PSP estimation, we first applied the algorithm to the CCH and ACHs of Fig. 3a–c and re-estimated the STG and PSP from the resulting dcCCH (Fig. 3g). Compared to the estimates derived from the CCH (Fig. 3d, e), all STG estimates based on the dcCCH were improved (Fig. 3h), and both PSP estimates were higher (Fig. 3i). Specifically, the tails predictor yielded an eSTG of 0.0408 (102% of the rSTG); the jitter predictor underestimated the rSTG (0.0276, 69%); and median filtering yielded an eSTG of 0.0392 (98%; Fig. 3h). We then carried out simulations in which the presynaptic burst fraction was varied in an orderly manner (from 0 to 0.4 with 0.05 increments; rSTG = 0.04; 30 repetitions each; other parameters were kept the same as in Fig. 3a–c). For each simulation, estimated STGs and PSPs were computed from the CCH and dcCCH using all five methods (Fig. 3j, k). For the tails, jitter, and median predictors, eSTGs derived from the dcCCH were more accurate than the eSTGs derived from the CCH when burst fractions were above 0.15 ($p < 0.001$, Mann–Whitney $U$-test). For the GLMCC method, only the PSP estimations derived from the raw CCH depended on the burst fraction (GLMCC without deconvolution: $p < 0.001$; with deconvolution: $p = 0.13$; permutation test). For comparing STG and PSP-based estimates, we used the normalized slope of the best linear fit, after dividing all values by the mean estimates at zero burst fraction. With the exception of CoNNECT, deconvolution reduced the effect of bursting for all methods ($p < 0.001$, bootstrap test; Fig. 3l). Thus, in the presence of burst spiking activity, deconvolution improves STG and PSP estimation of excitatory monosynaptic connections.

Second, to test whether deconvolution also improves the estimation of inhibitory monosynaptic connections, we repeated the simulations of Fig. 3g–l with inhibitory connections (rSTG = −0.02; Fig. 3m–r). The spike train of the presynaptic neuron realized a second-order gamma process modified by refractoriness ($\lambda_1 = 8$ spk/s; $ARP_1 = 2$ ms). The spike train of the postsynaptic neuron was simulated as a Poisson process modified by refractoriness ($\lambda_2 = 2$ spk/s; $ARP_2 = 2$ ms) which also exhibited bursting activity, varied in an orderly manner. For the tails, jitter, and median predictors, eSTGs derived from the dcCCH were more accurate than the eSTGs derived from the CCH when burst fractions were above 0.15 (Fig. 3p; $p < 0.001$, $U$-test). We then computed the normalized slope of the best linear fits (Fig. 3r). For all methods except CoNNECT, deconvolution reduced the effect of bursting (tails, jitter, median: $p < 0.001$; GLMCC: $p = 0.015$; bootstrap test). Thus, in the presence of burst spiking activity, deconvolution improves STG and PSP estimation for inhibitory monosynaptic connections.

**5**

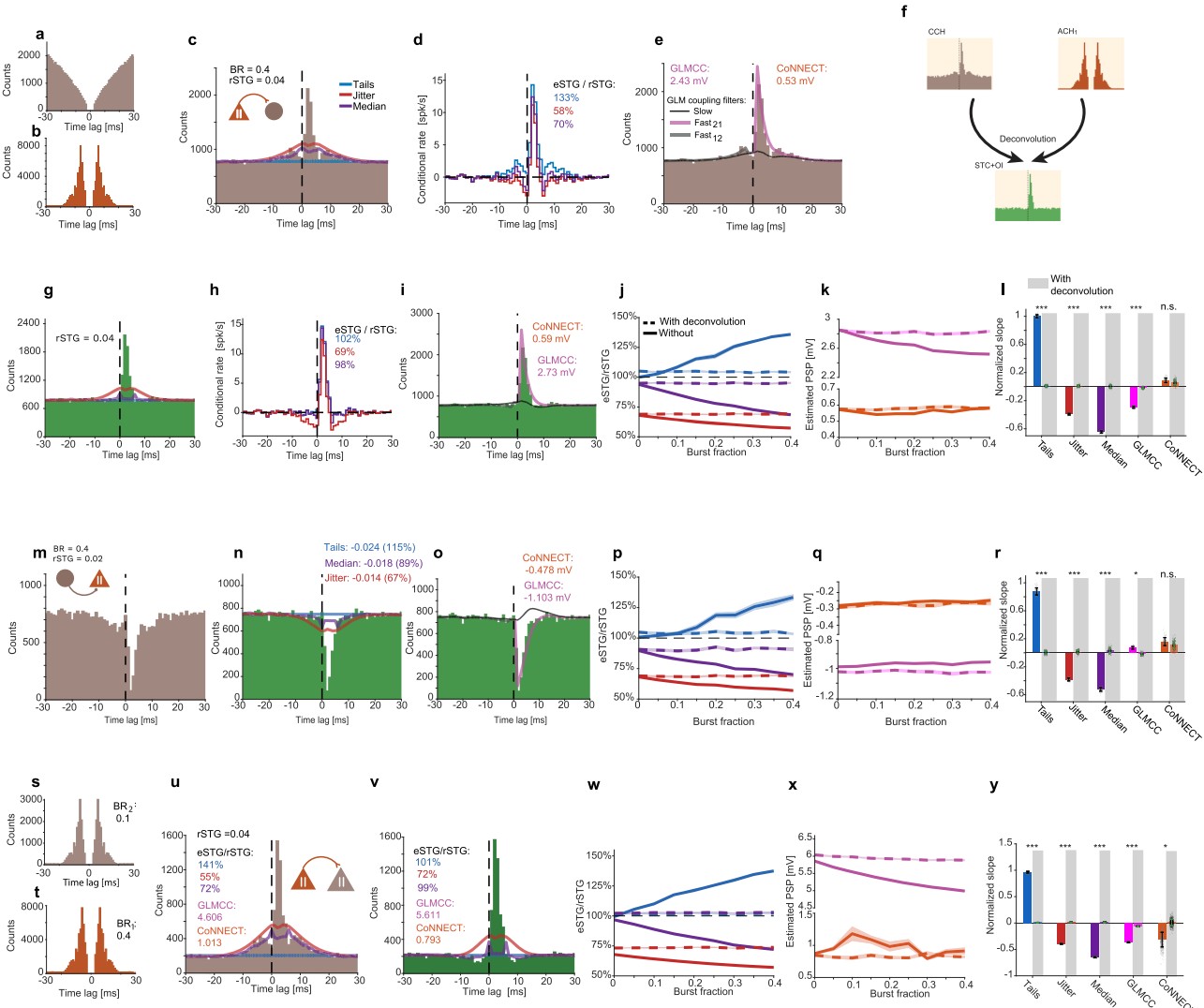

**Fig. 3 In the presence of spike bursts, deconvolution enables accurate recovery of spike transmission gain. a–e** Simulated spike trains of two neurons coupled by an excitatory monosynaptic connection. **a** ACH of the postsynaptic neuron. **b** ACH of the presynaptic neuron, exhibiting bursting activity. "Burst fraction" (BR; the probability of each spike to be first in a burst) is 0.4. **c** CCH between the two neurons. In addition to a peak within the causal ROI ($0 < \tau \leq 5$ ms), the CCH exhibits side lobes which are due to the burst spiking of the presynaptic neuron. **d** Conditional rate histograms, derived from the CCH in (**b**). **e** CCH with slow (black) and fast GLM coupling filters (pink and gray). **f** Burst spiking affects the ACH (**b**) and the CCH (**c**) and causes inaccurate STG estimates (**d**). The scheme illustrates the deconvolution process for removing ACH artifacts from the raw CCH. **g** A deconvolved CCH (dcCCH; green) is derived from the ACHs (**a**, **b**) and CCH (**c**). The dcCCH is free from the effect of spike bursts, recovering the unidirectional spike transmission curve $STG_{21}$ used in the simulation. **h** Conditional rate histograms. **i** dcCCH with PSPs estimated by the CoNNECT and GLMCC methods. **j**, **k** CCH deconvolution improves STG estimation in the presence of burst spiking. Spike trains coupled by an excitatory monosynaptic connection were simulated while modifying BR ($n = 30$ repetitions at $n = 9$ BR values, from 0 to 0.4 with 0.05 increments). **j** eSTG-to-rSTG ratio as a function of BR for the three STG estimation methods using CCHs (solid lines) and using dcCCHs (dashed lines). In the presence of bursting, deconvolution improves STG estimation. **k** Estimated PSP as a function of burstiness for the CoNNECT and GLMCC methods using CCHs and dcCCHs. **l** Normalized slope of the best linear fit of the curves in (**j**, **k**). Error bars, SEM. With the exception of CoNNECT, deconvolution reduces the effect of bursting for all methods ($n = 270$ random samples; n.s./***$p > 0.05/p < 0.001$, bootstrap test, $n = 300$ iterations). **m–o** Spike trains were coupled by an inhibitory monosynaptic connection. **m** Example CCH. **n** dcCCH for the same example. **o** dcCCH with slow GLM (black) and fast GLM coupling filters (pink and gray). **p**, **q** Same as (**j**, **k**), for simulated inhibitory connections ($n = 30$ repetitions at every BR value). **r** Same as (**l**), for inhibitory connections ($n = 270$ random samples; $n = 300$ bootstrap iterations). **s–v** Two spike trains with bursts were coupled by an excitatory monosynaptic connection. **s** ACH of the postsynaptic neuron (BR = 0.1). **t** ACH of postsynaptic neuron (BR = 0.4). **u** CCH of the two neurons. **v** dcCCH for the CCH in (**u**). **w**, **x** Same as (**j**, **k**), for two spike trains with bursts ($n = 30$ repetitions at every BR value). **y** Same as (**l**), for two spike trains with bursts ($n = 270$ random samples; $n = 300$ bootstrap iterations).

Third, to test whether deconvolution improves the estimation in the case of two directly-connected neurons which both exhibit bursty behavior, we simulated two bursting neurons coupled by a monosynaptic excitatory connection (rSTG = 0.04; Fig. 3s–u). The burst fraction of the postsynaptic neuron was 0.1 (Fig. 3s), and the burst fraction of the presynaptic neuron was varied from

0 to 0.4 with 0.05 increments. For every simulation, we estimated STGs and PSPs from the CCH and dcCCH using all five methods (Fig. 3u–x). We found that for the tails, jitter, and median predictors, eSTGs derived from the dcCCH were more accurate than eSTGs derived from the CCH for burst fractions above 0.1 ($p < 0.001$, $U$-test). For the CoNNECT method, only the ePSPs

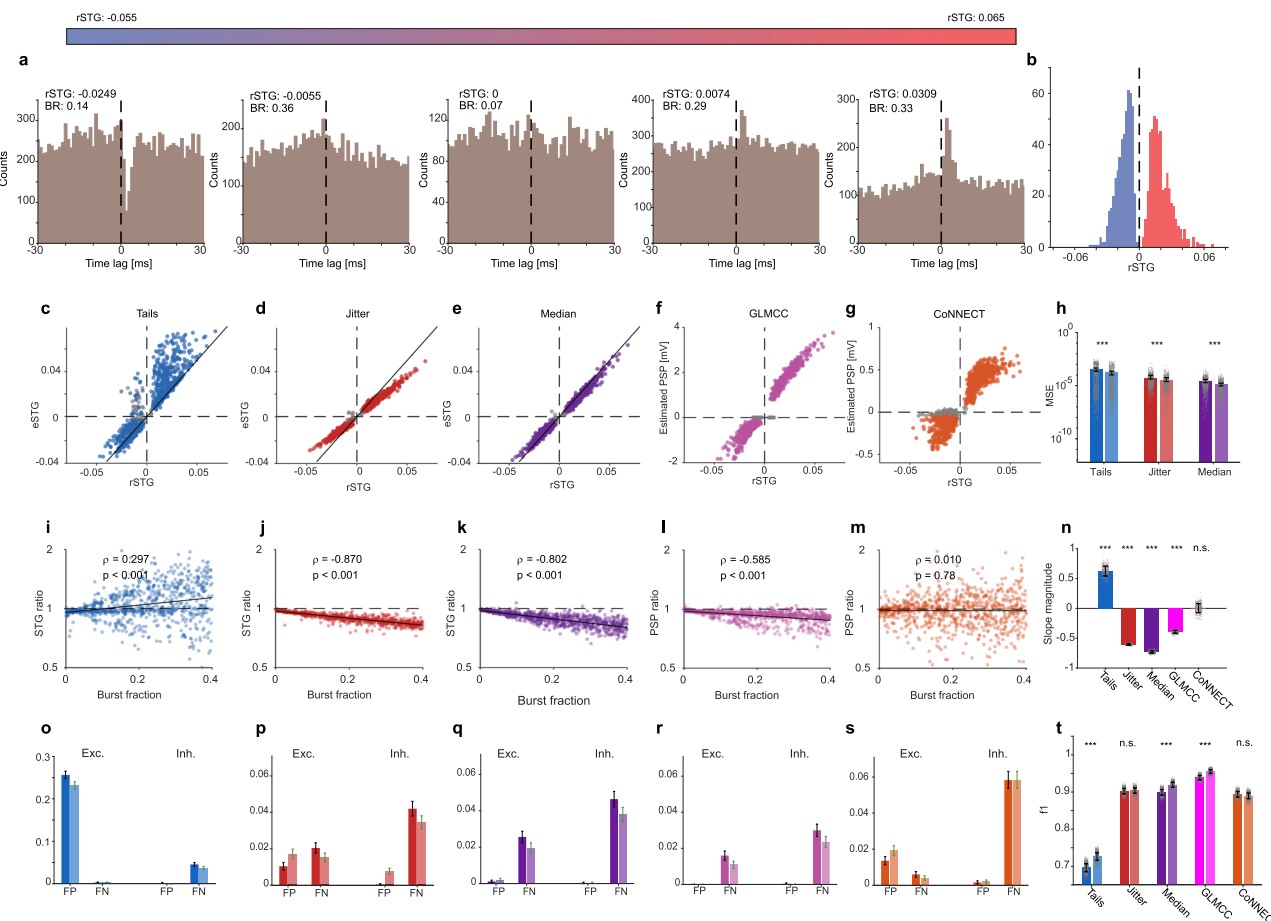

**Fig. 4 Deconvolution improves quantification and detection of inter-neuronal connections. a, b** A thousand pairs of neurons with monosynaptic connections and 250 unconnected pairs were simulated. Connectivity strengths (rSTG), burst fractions, co-modulation, and recording durations were varied between pairs. **a** Example CCHs from the noisy point process simulations, sorted by rSTG values. BR, burst fraction. **b** Empirical distribution of the $n = 500$ excitatory (red) and $n = 500$ inhibitory (blue) rSTGs of the pairs used in the simulations. **c–g** STGs or PSPs are plotted against the ground truth (rSTG) for dcCCHs ($n = 1000$ connected pairs), estimated using the tails (**c**), jitter (**d**), median (**e**), GLMCC (**f**), and CoNNECT (**g**) methods. Colored (or gray) dots depict estimated STGs or PSPs which were (or were not) detected as connections. In all methods (except CoNNECT), the significance threshold used was $\alpha = 0.001$. In (**c–e**), solid black lines represent perfect estimation (i.e., eSTG = rSTG). **h** Mean square errors (MSE) for the STG-based methods, in which the ground truth and the estimates have the same units. Error bars, SEM. Deconvolution reduces the MSE for all STG-based methods ($n = 1000$ connected pairs; ***$p < 0.001$, Wilcoxon's paired signed-rank test). The smallest errors are obtained for median filtering with deconvolution. **i–m** STG (**i–k**; or PSP, **l, m**) ratio as a function of burst fraction. STG ratio is defined as the STG estimated from the CCH, divided by the STG estimated from the dcCCH. The PSP ratio is defined in an analogous manner. For all methods except CoNNECT, deconvolution reduces the effect of burstiness. **n** Slope of best linear fit, computed for (**i–m**) ($n = 1000$ connected pairs; n.s./***$p > 0.05/p < 0.001$, bootstrap test, $n = 300$ iterations). **o–t** Effect of deconvolution on detection. **o–s** False-positive (FP) and false-negative (FN) rates were computed for excitatory (Exc.) and inhibitory (Inh.) connections, using CCHs (dark bars) and dcCCHs (light bars). Fractions are out of a population of {Exc. FP: $n = 2000$; Exc. FN: $n = 500$; Inh. FP: $n = 2000$; Inh. FN: $n = 500$} connections. **t** Mean $f_1$ scores for all connections, with and without deconvolution. Error bars, SEM. Deconvolution improves detection performance for the tails, median, and GLMCC methods (n.s./***$p > 0.05/p < 0.001$, bootstrap test, $n = 3500$ iterations).

derived from the raw CCH depended on the burst fraction (without deconvolution: $p = 0.01$; with deconvolution: $p = 0.053$; permutation test). For comparing STG and PSP-based estimates, we used the normalized slope of the best linear fit, after dividing all values by the mean estimates at zero burst fraction. Deconvolution reduced the effect of bursting for all methods (CoNNECT: $p = 0.025$; all other methods: $p < 0.001$, bootstrap test; Fig. 3y). Thus, when both neurons exhibit burst spiking activity, the deconvolution improves STG and PSP estimation.

**Deconvolution improves effective connectivity estimation and detection.** To evaluate the contribution of deconvolution to the performance of the various methods in a more realistic, noisy scenario, we generated 1250 individual neuronal pairs. Connectivity strengths, burst fractions, co-modulation, and recording

durations were varied widely between pairs (Fig. 4a). A thousand pairs were generated with either an excitatory or inhibitory connection, and 250 pairs were unconnected. For the excitatory and inhibitory pairs, connection strengths were drawn randomly from a log-normal distribution[43–45] (mean ± SD excitatory rSTG: $0.019 \pm 0.01$; inhibitory: $-0.014 \pm 0.007$; Fig. 4b). Burst fractions were drawn randomly from a uniform distribution in the [0 0.4] range. Recording duration was set randomly between 90 min and five hours.

First, we compared estimation performance with and without deconvolution by computing the mean square error (MSE) for the STG-based methods, for which a ground truth exists (Fig. 4c–e). We found that deconvolution reduced the MSE for all STG-based methods ($p < 0.001$; Wilcoxon's paired signed-rank test; Fig. 4h). Of the three STG-based methods, median filtering with

deconvolution had the lowest MSE ($1.27 \times 10^{-5}$). Median without deconvolution and jitter with deconvolution had the second and third lowest MSEs, respectively (median: $2.73 \times 10^{-5}$; jitter with deconvolution: $3.48 \times 10^{-5}$; jitter: $5.34 \times 10^{-5}$). Thus, deconvolution reduced the errors approximately two-fold. To evaluate the effect of deconvolution for all five methods, we computed the estimated STG (or PSP) ratio for both excitatory and inhibitory connections. For the "tails" method, the eSTG ratio increased with burst fraction ($\rho = 0.297$, $p < 0.001$, permutation test; Fig. 4i). For the jitter, median, and GLMCC methods, the estimated STG (or PSP) ratio decreased as burst fraction increased ($p < 0.001$, permutation test; Fig. 4j–l). For all methods (except CoNNECT), the slope of the estimated STG (or PSP) ratio was different from zero ($p < 0.001$, bootstrap test; Fig. 4n). Together, the reduction of the MSE (Fig. 4h) and the correlation between estimated STG (or PSP) ratio and burst fraction (Fig. 4n) show that deconvolution reduces the effect of burst spiking and improves connectivity estimation.

Second, we compared detection performance for all methods (Fig. 4o–t). We used an $f_1$ score that combines the effect of false-positive (FP) and false-negative (FN) rates across multiple populations (excitatory and inhibitory connections; Fig. 4t). Both the median and GLMCC, when employed with $\alpha = 0.001$, yielded low FP rates for excitatory and inhibitory connections. For median filtering, the FP rate for excitatory connections with/without deconvolution was 0.002/0.0012; for inhibitory connections, the FP rates (with and without deconvolution) were 0.0004 (Fig. 4q). For the GLMCC method, the FP rates for the excitatory connections with and without deconvolution were zero; for inhibitory connections, deconvolution improved the FP rate from 0.0004 to zero (Fig. 4r). Overall, deconvolution improved detection performance for the tails, median, and GLMCC methods ($p < 0.001$, bootstrap test; Fig. 4t). Of all methods, GLMCC with deconvolution yielded the highest $f_1$ score (mean [SEM]: 0.955 [0.004]). GLMCC without deconvolution and median filtering with deconvolution yielded the second and third scores (GLMCC: 0.94 [0.005]; median with deconvolution: 0.92 [0.006]; median: 0.90 [0.007]). Thus, deconvolution improves connectivity estimation and detection even for noisy CCHs.

**Deconvolution improves connection quantification and detection in networks of conductance-based CA1 model neurons.** The noisy point process simulations (Fig. 4) contained well-defined ground truth rSTGs, and generated CCHs that exhibited complex features and multiple types of spike-to-spike interactions. However, the point process simulations were limited to two-neuron "networks" with feedforward connectivity and may not exhibit other features, produced in larger-scale networks with reciprocally-connected neurons. To examine the contribution of deconvolution in such settings, we conducted simulations of noisy networks of conductance-based CA1 model neurons. In these simulations, the excitatory neurons (E-cells) were modeled using spiking models that allow incorporating graded bursting behavior (Fig. 5a). Bursting was controlled by a "burst factor" (BF) parameter, set to be between $-42\,\text{mV}$ (lower bursting activity) and $-35\,\text{mV}$ (higher bursting activity). Inhibitory neurons (I-cells) did not exhibit spontaneous bursting. In each simulation, a hundred-neuron network was constructed by connecting 80 E-cells and 20 I-cells. Connectivity patterns were implemented as often assumed for CA1, with E-to-I, I-to-E, and I-to-I connections, but without E-to-E connections[46]. The E- and I-cell ACHs produced under these conditions (Fig. 5b, c) are similar to those observed for pyramidal cells (PYR) and interneurons (INT) in real CA1 data (Fig. 6).

To focus on the contribution of deconvolution, we chose to use a single timescale separation method from this point onward. Of the STG-based methods, median filtering yielded the best performance, with respect to both quantification (Fig. 4h) and detection (Fig. 4t). Run times were tested on synthetic spike trains with firing rates of 2 and 8 spk/s, recorded over 5 h. The median [IQR] runtime of median filtering was 0.65 [0.64 0.67] ms, three to four orders of magnitude faster than the runtime of PSP-based methods (GLMCC: 3.72 [2.88 4.55] s; CoNNECT: 1.62 [1.54 1.69] s). These runtimes and performance made median filtering our method of choice for evaluating the contribution of deconvolution in biophysical simulations (Fig. 5) and in real data (Fig. 6).

To investigate how burst spiking activity affects STG estimation of pairs embedded in a neuronal network and whether deconvolution can improve STG estimation, we first conducted fixed-strength simulations. Every E-cell was connected randomly to one I-cell with a fixed excitatory synaptic conductance ($\text{Gie} = 0.055\,\text{mS/cm}^2$), and every I-cell was connected randomly to one E-cell and to one other I-cell with a fixed inhibitory conductance ($\text{Gei} = 0.25\,\text{mS/cm}^2$; Fig. 5d). Burstiness was randomized. Although E-to-I connection strengths were the same for all pairs, median filtering without deconvolution yielded eSTGs which were highly affected by the BF ($\rho = -0.68$, $p < 0.001$, permutation test; Fig. 5e). In contrast, median filtering with deconvolution produced eSTGs that were not correlated with burstiness ($\rho = -0.18$, $p = 0.11$, permutation test; Fig. 5f). The ratio between eSTGs without and eSTGs with deconvolution was close to one when BF was low ($-42\,\text{mV}$) and gradually decreased as BF increased ($\rho = 0.891$, $p < 0.001$, permutation test, Fig. 5g). The eSTG ratio for the I-to-E connections showed a similar trend, in which the eSTG ratio was negatively correlated with the BF ($\rho = -0.835$, $p < 0.001$, permutation test; Fig. 5h). Thus, even when connections have fixed strengths, the eSTGs depend on burst spiking activity. However, dependency is minimized using deconvolution.

To evaluate the contribution of deconvolution in a more heterogenous setting, we generated networks of conductance-based CA1 model neurons with varied connectivity strengths (drawn randomly from log-normal distributions; Fig. 5i). Other parameters were the same as in Fig. 5d–h. Over six hundred-neuron networks, the mean [SEM] $f_1$ score obtained with deconvolution was 0.89 [0.008], higher than $f_1$ without deconvolution (0.84 [0.009]; $p < 0.001$, bootstrap test; Fig. 5j–m). In addition to improved detection, deconvolution yielded eSTG estimates which were higher than eSTG estimates without deconvolution, especially when bursting was prevalent (Fig. 5n–s). eSTG ratios for the E-to-I and I-to-E connections were negatively correlated with BF (E-to-I: $\rho = -0.88$, $p < 0.001$, permutation test, Fig. 5p; I-to-E: $\rho = -0.837$, $p < 0.001$; Fig. 5s). Thus, in synthetic neuronal networks, deconvolution improves both the detection and quantification of inter-neuronal connectivity.

**In real neuronal data, deconvolution-based estimates of spike transmission gain are especially higher when burst spiking is prevalent.** To test the contribution of deconvolution to the analysis of real neuronal data, we obtained data from freely-moving mice. The dataset consisted of 1041 PYR and 215 INT recorded using high-density silicon probes implanted in hippocampal region CA1 of three mice (Fig. 6a). In these data, the median [IQR] firing rates of PYR and INT were 0.90 [0.14 1.68] spk/s and 9.44 [1.04 17.83] spk/s, respectively (Fig. 6b). For quantifying burst spiking activity of real spike trains, we used the "burst index". Burst indices were 0.28 [0.1 0.47] for PYR and $-0.283$ [$-0.37$ $-0.19$] for INT (Fig. 6c). In contrast to simulations,

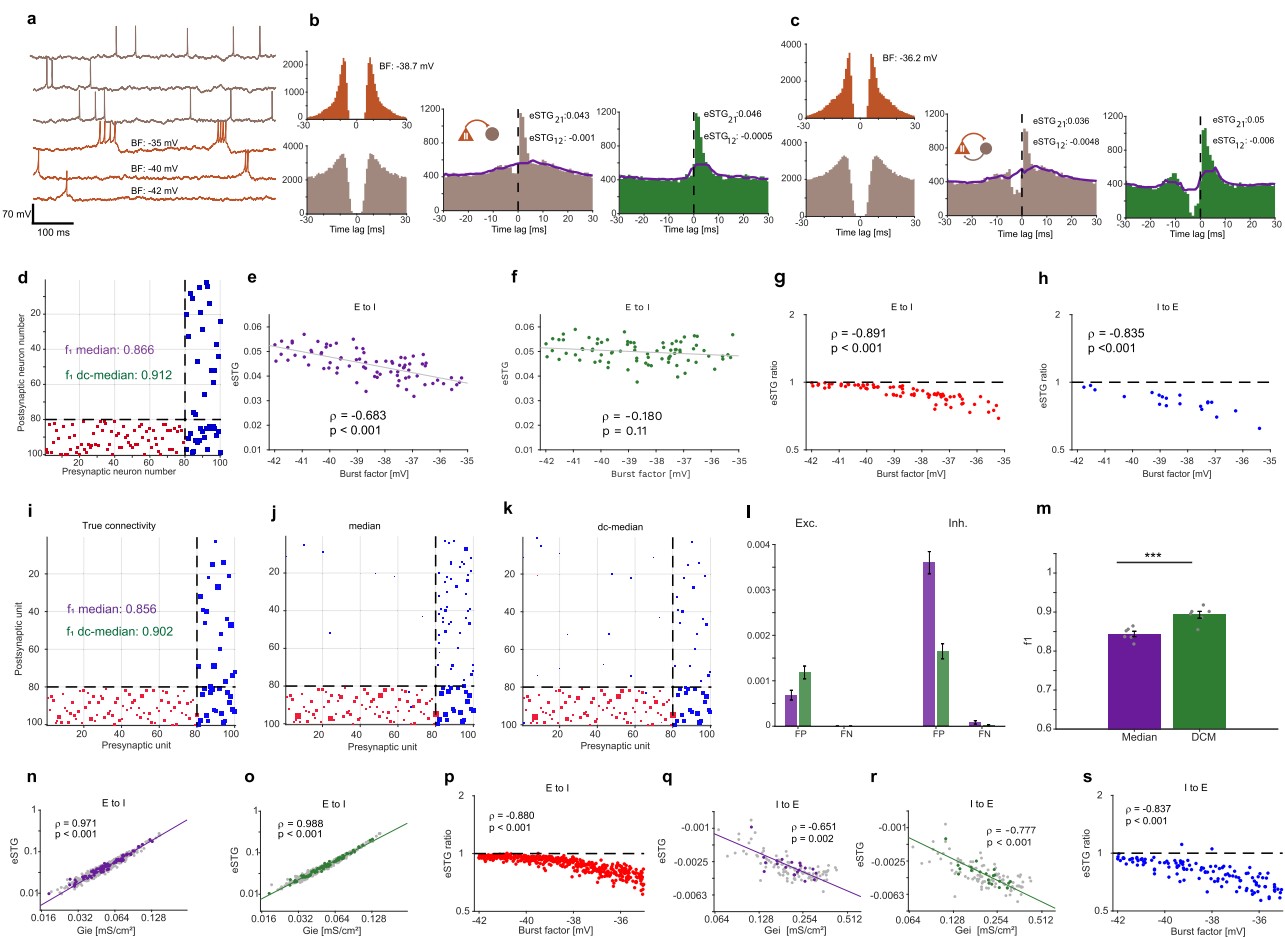

**Fig. 5 Deconvolution improves connection quantification and detection in networks of conductance-based CA1 model neurons. a** Traces show example membrane potentials of six simultaneously-simulated cells. Gray, inhibitory neurons (I-cells); orange, excitatory neurons (E-cells), sorted according to burstiness, quantified by the burst factor (BF). **b**, **c** Example ACHs and CCHs for two E-to-I cell pairs. **b** A pair with feedforward excitation ($Gie = 0.055$ mS/cm$^2$, $Gei = 0$ mS/cm$^2$). **c** A pair with reciprocal excitation and inhibition ($Gie = 0.055$ mS/cm$^2$, $Gei = 0.25$ mS/cm$^2$). Both pairs were embedded in the same hundred-cell network; in both cases, deconvolution increased the eSTGs. **d**–**h** A network of 100 neurons (80/20 E/I cells) with fixed connectivity strengths was simulated. E-cell BFs were drawn uniformly from the $[-42\ -35]$ mV range. **d** Hinton diagram of network connectivity. Red/blue squares represent excitatory/inhibitory connections, and square size is proportional to connectivity strength. Dashed lines separate E- and I-cells. **e**, **f** eSTGs of $n = 80$ fixed E-to-I connections vs. BF using median filtering without (**e**) and with (**f**) deconvolution. Only eSTGs estimated without deconvolution are correlated with the BF. **g**, **h** eSTG ratios for $n = 80$ E-to-I (**g**) and for $n = 20$ I-to-E (**h**) connections. **i**–**s** Six different hundred-neuron networks were simulated with synaptic conductance values drawn randomly from log-normal distributions. Other details were the same as in (**d**–**h**). **i**–**k** Hinton diagrams for one of the networks. **l** FP and FN rates were computed for all datasets using median filtering without (purple bars) and with deconvolution (green bars). Fractions are out of a population of {Exc. FP: $n = 59,520$; Exc. FN: $n = 480$; Inh. FP: $n = 59,760$; Inh. FN: $n = 240$} connections. Error bars depict SEM. **m** Mean $f_1$ scores for all datasets and connection types. Error bars, SEM. $n = 6$ networks. $p < 0.001$, bootstrap test, $n = 3500$ iterations. **n**, **o** eSTGs for E-to-I connections plotted against true connectivity magnitude (the excitatory conductance, $Gie$). Colored dots depict $n = 80$ connections from the dataset shown in (**i**), and gray dots depict $n = 400$ connections from other datasets. The best linear fit (solid line) and Spearman's correlation coefficient are shown for the colored dots. **p** eSTG ratios for $n = 480$ E-to-I connections of all datasets. eSTG ratios are negatively correlated with BF. **q**, **r** Same as **n**, **o**, for I-to-E connections. Colored dots depict $n = 20$ connections from the dataset shown in (**i**), and gray dots depict $n = 100$ connections from other datasets. **s** eSTG ratios for $n = 120$ I-to-E connections of all datasets. Ratios are negatively correlated with BF.

rSTGs are of course unknown for the real neuronal data. Therefore, we tested the contribution of deconvolution by measuring the eSTG ratio (without and with deconvolution) as a function of the burst index. Only PYR-to-INT pairs with a consistent monosynaptic CCH peak ($p < 0.001$, Poisson test), in which the PYR exhibited a non-negative burst index, were analyzed (1274/7991 pairs). For the analysis pairs, firing rates of PYR and INT were 0.96 [0.23 1.7] spk/s and 13.05 [5.27 20.91] spk/s, respectively (Fig. 6b). Burst indices were 0.33 [0.18 0.49] for PYR and $-0.28$ [$-0.348\ -0.219$] for INT (Fig. 6c). For the analysis pairs only, deconvolution-based eSTGs were consistently higher than eSTGs without deconvolution (medians: 0.019 and 0.017, respectively; $p < 0.001$, Wilcoxon's paired signed-rank; Fig. 6d).

Thus, in pairs of real neurons with putative monosynaptic connections, deconvolution yields higher eSTGs than eSTGs estimated without deconvolution.

The higher eSTGs yielded by median filtering with deconvolution may result from a fixed DC shift or be burst-dependent. To differentiate between these possibilities, we measured eSTG ratios as a function of PYR burst index. To visualize the effect of deconvolution, we chose several pairs of units (Fig. 6e, f) in which the PYR exhibited burst spiking activity (Fig. 6e, insets). The CCHs between the units exhibited prominent peaks within the causal ROI ($0 < \tau \leq 5$ ms), consistent with monosynaptic connectivity. In addition, the CCHs exhibited side lobes on both sides of the main peak, consistent with PYR bursts (Fig. 6e). In

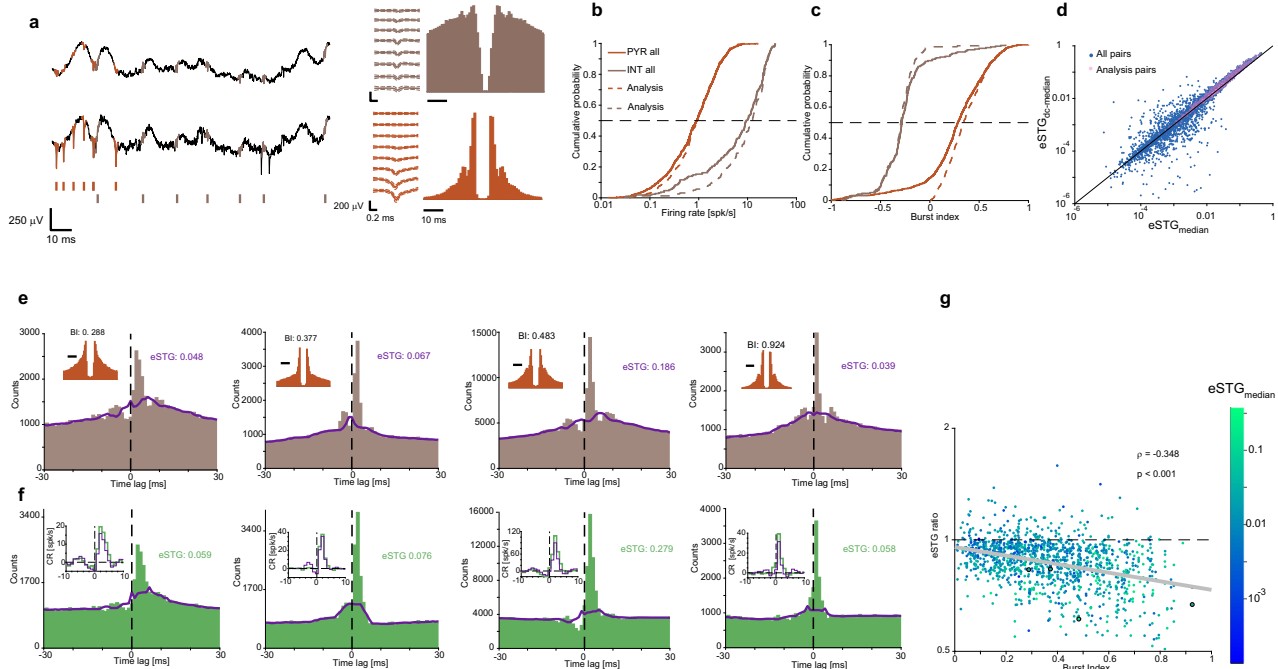

**Fig. 6 In real neuronal data, deconvolution-based estimates of spike transmission gain are especially higher in the presence of burst spiking activity.** **a**–**c** Data were recorded from hippocampal region CA1 of three freely-moving mice using high-density silicon probes. **a** Example wideband data (band-pass filtered 0.1–7500 Hz) recorded by two sites on the same shank, separated vertically by 60 μm. Orange and brown traces and tick marks correspond to PYR and INT spikes, respectively. **b** Cumulative distributions of the firing rates of $n = 1041$ PYR and $n = 215$ INT (solid lines). Dashed lines show distributions of $n = 611$ PYR and $n = 152$ INT used for further analyses (**d**, **e**). **c** Distributions of burst indices of the same four neuronal groups. **d** In PYR-to-INT pairs with putative monosynaptic connections, STG estimates based on deconvolution are higher than estimates that employ only median filtering. Scatter plot shows $eSTG_{dc-median}$ vs. $eSTG_{median}$ for all $n = 7991$ PYR-to-INT pairs in which both eSTGs were positive. PYR-to-INT pairs with a consistent monosynaptic CCH peak, in which the PYR also exhibits a non-negative burst index, are denoted as "analysis pairs" (pink dots; $n = 1274$ pairs). For the analysis pairs, the $eSTG_{dc-median}$ is higher than the $eSTG_{median}$ ($p < 0.001$, Wilcoxon's paired signed-rank test). **e** Four example CCHs with the median predictor. Insets: The ACHs of the putative presynaptic neurons, which exhibit burst spiking. **f** dcCCHs for the pairs in (**e**). Insets: Conditional rate (CR) histograms based on the CCHs (purple) and the dcCCH (green). In all examples, the deconvolution-based eSTG is higher than the eSTG estimated without deconvolution. **g** eSTG ratios for the full dataset ($n = 1274$ pairs), shown as a function of the presynaptic burst index. Deconvolution-based estimates are consistently higher than estimates without deconvolution ($p < 0.001$, Wilcoxon's signed-rank test). Estimates are particularly higher when spike bursting is more prevalent ($p < 0.001$, permutation test).

contrast, the dcCCHs did not exhibit side lobes (Fig. 6f), and deconvolution-based eSTGs were higher than eSTGs estimated from the CCHs. Over 1274 pairs, eSTG ratios exhibited a negative correlation with burst indices ($\rho = -0.348$; $p < 0.001$, permutation test; Fig. 6g). Furthermore, most eSTG ratios were smaller than one (median: 0.88; $p < 0.001$, Wilcoxon's signed-rank test; Fig. 6g). The negative correlation between eSTG ratio and burst index in real neuronal data, and the fact that most eSTG ratios are smaller than one, are consistent with the controlled simulations (Figs. 3–5). Together, these observations suggest that deconvolution improves both the quantification and the detection of inter-neuronal connections.

## Discussion
We showed that the CCH between two neurons can be decomposed into a sum of three elements. Based on the mathematical formulation, we developed a deconvolution-based algorithm that removes second-order spike train statistics from the CCH, improving connectivity quantification and detection. Using pairwise point process and conductance-based network simulations, we showed that burst spiking activity modifies the CCH, impairing connectivity estimates yielded by several distinct methods. Deconvolution removed the effect of burst spiking from the CCH and recovered the constructed connectivity. In noisy point process simulations with known STGs, deconvolution

reduced the errors of all STG-based methods, and improved detection performance for the tails, median, and GLMCC methods. In networks of conductance-based CA1 model neurons, the eSTGs yielded by median filtering depended on burst spiking activity, and this dependency was removed by deconvolution. In real neuronal data recorded from CA1 region of freely-moving mice, deconvolution-based eSTGs were consistently higher than eSTGs without deconvolution, in particular when bursting was prevalent. Together, the results imply increased accuracy upon using deconvolution when analyzing real neuronal data.

Estimates of STG/PSP yielded by all methods employed (tails, jitter, median, GLMCC, and CoNNECT) were averaged over time and spike patterns. The underlying assumption was that the ACHs and connectivity strengths are fixed over time. However, previous studies have shown that the postsynaptic effect of a presynaptic spike may depend on the presynaptic inter-spike interval[3,10,11] and on other parameters[32,47]. In vitro work showed that synaptic transmission can exhibit short-term plasticity, where the EPSP magnitude in a postsynaptic neuron is modified according to the order and timing of presynaptic spikes[48]. In some cases, EPSP magnitude decreased over consecutive spikes, and in other cases consecutive spikes caused facilitation of EPSP magnitude[3,49]. Consistent with the in vitro work, spike transmission estimates derived from the CCH in vivo were distinct when CCHs were computed using the first, second, or third spike in a presynaptic burst[21,32,50,51]. Other in vivo studies used trial-

averaging and time-resolved cross-correlation measures, showing that spike-to-spike correlations depend on the time lag relative to a stimulus or an action[32,47,52]. As is, the deconvolution approach presented here (as well as the timescale separation methods employed) does not take into account time-dependent changes in the auto-correlation (e.g., of bursting activity) and cannot estimate time-dependent changes of spike transmission. However, deconvolution together with a timescale separation method can be applied to segmented data to track time-dependent changes in spike transmission (as done in refs. [21,32,47,51,52]), yielding estimates differentiated from synchronous firing and burst spiking activity.

We showed that deconvolution improves connectivity estimates for bursting activity, but deconvolution can remove other effects of second-order spike train statistics on the CCH. While bursting is a relatively common deviation from Poisson spiking in real spike trains, other deviations are clearly possible. For instance, in some brain regions, strong periodicity is encountered, for instance at the ripple[53] or gamma[54] range. When individual neurons exhibit phase locking to field oscillations, the ACH may exhibit periodicity[54,55]. Then, the CCH between connected neurons will be affected by ACH periodicity. Deconvolution can be used to differentiate between the effect of transmitted spikes and the firing patterns of the presynaptic neuron. Thus, deconvolution can minimize the distortion of connectivity estimates and improve the analysis of slower CCH features.

Following multiple technological advances, the capability to record data from many neuronal pairs has drastically increased. The temporal resolution of calcium imaging techniques has gradually improved[56–60]. Furthermore, voltage imaging now allows recording the activity of multiple neurons at sub-millisecond temporal resolutions[61,62]. Large-scale electrophysiological techniques have evolved to enable recording hundreds of spike trains from the same or different brain areas simultaneously over prolonged durations[63–65]. Analyzing neural circuit dynamics based on massively parallel spike timing data requires adequate quantification of synaptic connectivity. In recent years, several analysis methods have been introduced, producing increasingly more accurate circuit maps based on multiple spike train recordings. Deconvolution can improve the outcome of those methods, and has the potential to improve the performance of any CCH-based method. The accuracy and sensitivity improved by deconvolution enables the construction of more accurate neuronal connectivity maps which may include weaker or otherwise hidden connections. Detailed maps, which take into account overlooked connectivity motifs, may enhance our understanding of the neural mechanisms underlying brain function.

## Methods

### Conductance-based model of a leaky integrate and fire (LIF) neuron driven by a Poisson train

To quantify the interplay between spike transmission gain (STG), unitary EPSP (uEPSP) amplitude, and background activity we constructed a simple conductance-based, leaky integrate and fire (LIF) model neuron with membrane potential $V_2$ driven by a Poisson spike train via a single excitatory synapse (LIF-syn model). The model was governed by two differential equations and a single condition:

$$C\frac{dV_2(t)}{dt} = I(t) - g_L(V_2(t) - E_L) - g_S S(t)(V_2(t) - E_S) + g_N \eta(t) \quad (4.1)$$

$$\text{if } V_2 \geq V_{th}, V_2 \leftarrow V_{reset} \quad (4.2)$$

$$\frac{dS(t)}{dt} = H(V_1)(1 - S(t))/\tau_r - S(t)/\tau_d \quad (4.3)$$

We used $C = 1$ μF/cm², $g_L = 0.5$ mS/cm², $E_L = -60$ mV, $g_S \in [0\ 0.032]$ mS/cm², $E_S = 0$ mV, $g_N = 1$ mS/cm², $V_{th} = -50$ mV, $V_{reset} = -70$ mV, $\tau_r = 0.1$ ms, and $\tau_d = 3$ ms. Membrane potential variability of the postsynaptic neuron, which may stem from many unknown sources, was quantified in the LIF model by additive Gaussian noise $\eta(t) \sim G(0, \sigma)$, with an SD that was varied between runs, $\sigma \in [0\ 5]$ mV.

Whenever a LIF spike occurred, $V_2$ was held at $V_{peak} = 50$ mV for $T_{spike} = 1$ ms before being reset to $V_{reset}$ for another $ARP_2 - T_{spike} = 1$ ms.

Explicit external inputs to the model were a fixed bias current, $I(t) = 1$ μA/cm² $\forall t$, and a presynaptic spike train. The presynaptic spike train was generated as a homogeneous Poisson process with $\lambda_1 = 5$ spk/s, and then spikes were decimated to maintain a minimal inter-spike interval of 4 ms. Whenever a presynaptic spike occurred, $V_1 = V_{peak} = 50$ mV for $T_{spike} = 1$ ms; otherwise $V_1 = E_L = -60$ mV. For incorporating the presynaptic spike train in the synaptic differential equation, we used $H(V) = (1 + \tanh(V/4))/2$. Thus, whenever an isolated presynaptic spike occurred, the synaptic variable $S$ rose exponentially (with $\tau_r$) toward one over 1 ms, and then relaxed back to zero (with $\tau_d$). Short-term synaptic plasticity (facilitation and/or depression) was not included. The model was implemented using explicit second-order Runge-Kutta endpoint numerical integration method with a time step of $\Delta t = 0.1$ ms and total duration of $T_{sim} = 180$ min.

For constructing Fig. 1a, the LIF did not receive any explicit synaptic input ($g_S = 0$ mS/cm²), but $\sigma \in [0\ 5]$ mV was varied at 0.5 mV steps (11 levels). For every $\sigma$ level, six repetitions were performed. For constructing Fig. 1b, we also employed 11 levels of $g_S$, matched to yield uEPSP amplitudes $uEPSP \in [0\ 2]$ mV at 0.2 mV steps. For every $\{\sigma, uEPSP\}$ combination, six repetitions were performed.

### Quantification of spike transmission gain in the LIF-syn model

The firing rate of the LIF, $\lambda_2$, was determined by counting all spikes and dividing by $T_{sim}$. Denoting the firing rate of the Poisson input by $\lambda_1$, the "real" spike transmission gain (rSTG) in the stationary scenario may be intuitively defined as the number of spikes added due to the presence of a synapse, divided by the number of presynaptic spikes, namely $\triangle\lambda_2/\lambda_1 = (\lambda_2^{syn} - \lambda_2^{no-syn})/\lambda_1$. However, such computation is flawed since the addition of transmitted spikes occasionally decimates spontaneous (noise-induced) spikes. In the case of a LIF model with an absolute refractory period $ARP_2$ and time step $\Delta t$, the exact rSTG is obtained by multiplying the naive ratio by a correction factor that scales the firing rate difference by the fraction of non-occupied bins:

$$\text{rSTG} = \frac{\lambda_2^{syn} - \lambda_2^{no-syn}}{\lambda_1}\left(1 - \lambda_2^{no-syn} \cdot (ARP_2 + \Delta t)\right). \quad (5)$$

### Spike cross-correlation histograms

Denote two spike trains by $s_1$ and $s_2$. Each train is a sum of delta functions recorded over a duration $T$ and can be expressed as $s_1(t) = \sum_{i=1}^{N_1}\delta(t - t_i), \forall 0 \leq t_i \leq T$ and $s_2(t) = \sum_{i=1}^{N_2}\delta(t - t_i), \forall 0 \leq t_i \leq T$, respectively. Then, the cross-correlation between the two trains at any time lag $\tau$ is $CC(\tau) = \int_0^\infty s_1(t)s_2(t + \tau)dt$. Equivalently, the spike trains can be expressed as ordered lists: $s_1 = \{t_i; i = 1, \dots, N_1\}, \forall 0 \leq t_i \leq T$; and $s_2 = \{r_i; i = 1, \dots, N_2\}, \forall 0 \leq r_i \leq T$. Then, the cross-correlation can be expressed as $CC(\tau) = \sum_{i=1}^{N_1}\sum_{j=1}^{N_2}\#\left\{(t_i - r_j) = \tau\right\}$, where the indicator function is defined as $\#\{true\} = 1$ and $\#\{false\} = 0$. In practice, two spikes rarely occur at the exact same instant, and the cross-correlation is expressed as a histogram with a finite bin size $B$ and a total of $2M + 1$ bins, spanning the time range from $-MB$ to $MB$. Then, the CCH count at the $m$th bin ($m \in [-M, -M + 1, \dots 0, \dots, M - 1, M]$) sums the cross-correlation over the time range $(m - 1/2)B \leq \tau < (m + 1/2)B$, and is expressed as $CCH[m] = \sum_{i=1}^{N_1}\sum_{j=1}^{N_2}\#\left\{(m - 1/2)B \leq (t_i - r_j) < (m + 1/2)B\right\}$. Unless mentioned otherwise, we used a bin size of $B = 1$ ms and histogram range of $2M + 1 = 61$ ms. We used a bin size of 1 ms to minimize the probability that two spikes of the same train will occur in the same bin, keeping the counting process Poisson. A 1 ms bin size has used in multiple extracellular[22], intracellular[2], and across species[66] studies. Furthermore, the CoNNECT method can operate only on CCH with bin size of 1 ms.

### "Other inputs" predictor

In this work we used four methods to estimate the "other inputs" element of the CCH: the tails predictor, the jitter predictor, the median filter predictor, and the slow part of a generalized linear model for cross-correlations (GLMCC). The tails predictor is computed by averaging over all CCH bins that are far from the zero-lag bin, both causal and anti-causal:

$$\text{pred}_{tails}[m] = \frac{1}{K}\sum_k CCH[k], k = \{-M, -M + 1, \dots, -M_0, M_0, \dots, M - 1, M\} \quad (6.1)$$

The sum is over all $K$ bins for which $|k| \geq M_0$; we used $M_0 = 11$ ms. Therefore, the tails predictor has the same value for all $2M + 1$ CCH bins (a straight line). The jitter predictor is computed by convolving the CCH with a partially hollowed Gaussian kernel:

$$\text{pred}_{jitter}[m] = \sum_{k=-3\delta}^{3\delta} CCH[m - k]w[k] \quad (6.2)$$

The Gaussian kernel, denoted as $w[k]$, has a unity sum, standard deviation $\delta$, support of $6\delta$, and the central bin is partially hollowed (0.6). Unless noted otherwise, we used $\delta = 5$ ms. Jittering every spike within a rectangular $\pm\delta$ window centered around the spike ("spike time jitter") many times and computing an average jittered CCH converges exactly to the convolution of the CCH with a

triangular window that has a width of $4\delta + 1$[20]. Convolution with a partially hollowed Gaussian window with an SD of $\delta$ reduces the false-negative rate and increases detection power[20]. The hollowed median filter predictor is computed by applying a median filter of order $2\delta$ over the CCH, while ignoring the value in the central bin:

$$\text{pred}_{\text{median}}[m] = \text{median}(\text{CCH}[m - \delta, \dots, m - 1, m + 1, \dots m + \delta]) \quad (6.3)$$

The GLMCC predictor is given by the slow part, $a(t)$, of the GLM defined by refs. [40,41].

**Conditional rate CCH and spike transmission curve**. To estimate the spike transmission curve (STC) from the CCH, we first compute the conditional rate CCH (crCCH). The crCCH is derived by subtracting, for every time lag, the predictor from the CCH, and scaling by the number of trigger spikes and the bin size:

$$\text{crCCH}[m] = (\text{CCH}[m] - \text{pred}[m])/(N_1 \cdot B) \quad (7)$$

Thus, the crCCH has units of spikes per second (spk/s). Second, we find the extremum value of the crCCH within the casual temporal region of interest (ROI: $0 < m \le 5$ ms). The two zero-crossing points, to the left and to the right of the extremum value, are defined as bounds, $B_L$ and $B_R$. To preserve causality, if $B_L$ falls at the zero or a negative time lag, the first bin after the zero-lag bin is used instead. However, $B_R$ may be outside (to the right of) the ROI. The STC is then defined as equal to the crCCH for all bins within the $[B_L, B_R]$ interval, and zero otherwise (Fig. 1c).

The ROI can be modified according to the expected form of the STC, based on two parameters: the delay and the jitter of spike transmission. When the transmission delay is short and the STC is narrow, a short ROI near-zero lag is suitable. Here, an ROI of (0,5) was chosen to ensure including at least part of the monosynaptic STC, consistent with previous results in neocortex[32] and hippocampus exhibiting a transmission delay in the 0–3 ms range[21,22,67,68].

**Computation of spike transmission gain**. The estimated spike transmission gain (eSTG) is the integral over the STC, computed numerically as the sum of all estimated STC values multiplied by the bin size:

$$\text{eSTG} = \sum_{m=B_L}^{B_R} \text{STC}[m] \cdot B \quad (8)$$

**Mathematical framework for deconvolved CCH**. The deconvolution approach is based on the realization that the CCH between two spike trains can be expressed as an exact sum of four elements and approximated by a sum of three elements. The decomposition (Eq. 17) is mathematically exact when the system is assumed to be linear time invariant (LTI). "Linear" implies that the spike transmission process is linear, but does not imply linearity of spike generation, which is a highly nonlinear process. Linearity of spike transmission means that the process of transmitting spikes is additive and does not depend on the history of spike transmission. Linearity is incompatible with neuronal refractoriness, and is assumed for mathematical simplicity. "Time invariant" implies that the impulse responses of spike transmission—and in this specific case, also the ACHs—are constant over the duration of the recording. These assumptions are implicit in the computation of any CCH and in most CCH applications (see "Discussion").

Within the LTI framework, we proceed as follows. The specific system studied is a pair of reciprocally-coupled neurons, each of which also receives external input (Fig. 2a). Formally, the observed spike train of neuron 1, $s_1$, is a sum of delta functions and thus has the form

$$s_1(t) = \sum_{i=1}^{N_1} \delta(t - t_i), \forall 0 \le t_i \le T \quad (9)$$

$N_1$ is the total number of spikes in $s_1$, $t_i$ is the instant at which the $i$th spike occurred, and $T$ is the total recording duration. The spike train of neuron 2, $s_2$, is defined in an analogous manner by $s_2(t) = \sum_{i=1}^{N_2} \delta(t - t_i)$, $\forall 0 \le t_i \le T$. We denote the finite impulse response of spike transmission from neuron 1 to neuron 2 by

$$h_1(\tau), \forall 0 < \tau \le \tau_{\max} \quad (10.1)$$

The impulse response between neuron 2 and neuron 1, $h_2$, is denoted in an analogous manner by

$$h_2(\tau), \forall 0 < \tau \le \tau_{\max} \quad (10.2)$$

Specifically, $h_1$ and $h_2$ are non-zero only for positive time lags, which is the standard definition of causality. Using these definitions, the spike train of neuron 2, $s_2$, is the superposition of spikes from two sources: (i) transmitted spikes, due to $s_1$ filtered by the synaptic connection between the two neurons, $h_1$; and (ii) background spikes, denoted by $s_2^0$, which are due to other sources:

$$s_2(t) = s_2^0(t) + \int_{-\infty}^{\infty} s_1(t - \tau)h_1(\tau)d\tau \quad (11)$$

The spike train of neuron 1, $s_1$, can be expressed in an analogous manner. To facilitate the analysis, we take the Fourier transform of all quantities, and obtain the frequency domain representation of Eq. (11):

$$S_2(f) = S_2^0(f) + S_1(f)H_1(f), \quad (12.1)$$

$S_2(f)$ is the Fourier transform of $s_2(t)$, $S_2(f) = \int_{-\infty}^{\infty} s_2(t)\exp(-2\pi i f t)dt$. In an analogous manner, we obtain the Fourier transform of $s_1$:

$$S_1(f) = S_1^0(f) + S_2(f)H_2(f) \quad (12.2)$$

Equation 12 consists of a pair of coupled equations, for which the solution is:

$$S_1(f) = \frac{S_1^0(f) + S_2^0(f)H_2(f)}{1 - H_1(f)H_2(f)} \quad (13.1)$$

$$S_2(f) = \frac{S_2^0(f) + S_1^0(f)H_1(f)}{1 - H_1(f)H_2(f)} \quad (13.2)$$

To link the CCH with the impulse response, the cross-correlation between the two spike trains

$$\text{CCH}_{12}(\tau) = \int_{-\infty}^{\infty} s_1(t)s_2(t + \tau)dt \quad (14.1)$$

is written in the frequency domain as

$$\mathscr{F}(\text{CCH}_{12}) = \overline{S_1(f)}S_2(f) \quad (14.2)$$

where $\overline{S_1(f)}$ denotes complex conjugation, due the fact that when computing convolutions—but not cross-correlations—one of the signals is time reversed. Plugging Eq. (13) into Eq. (14.2) we obtain

$$\mathscr{F}(\text{CCH}_{12}) = \frac{\overline{S_1^0(f)}S_2^0(f) + \overline{S_1^0(f)}S_1^0(f)H_1(f) + \overline{S_2^0(f)}S_2^0(f)\overline{H_2(f)} + \overline{S_2^0(f)}S_1^0(f)\overline{H_2(f)}H_1(f)}{\left|1 - H_1(f)H_2(f)\right|^2} \quad (15)$$

The elements in the numerator of Eq. (15) can be rewritten in terms of correlation functions of the latent (background) spike trains as follows:

$$\text{CCH}_{12}^0(\tau) = \overline{S_1^0(f)}S_2^0(f) = \int_{-\infty}^{\infty} s_1^0(t)s_2^0(t + \tau)dt \quad (16.1)$$

$$\text{CCH}_{21}^0(\tau) = \overline{S_2^0(f)}S_1^0(f) = \int_{-\infty}^{\infty} s_2^0(t)s_1^0(t + \tau)dt \quad (16.2)$$

$$\text{ACH}_1^0(\tau) = \overline{S_1^0(f)}S_1^0(f) = \int_{-\infty}^{\infty} s_1^0(t)s_1^0(t + \tau)dt \quad (16.3)$$

$$\text{ACH}_2^0(\tau) = \overline{S_2^0(f)}S_2^0(f) = \int_{-\infty}^{\infty} s_2^0(t)s_2^0(t + \tau)dt \quad (16.4)$$

Yielding

$$\mathscr{F}(\text{CCH}_{12}) = \frac{\mathscr{F}(\text{CCH}_{12}^0) + \mathscr{F}(\text{ACH}_1^0)H_1(f) + \mathscr{F}(\text{ACH}_2^0)\overline{H_2(f)} + \mathscr{F}(\text{CCH}_{21}^0)\overline{H_2(f)}H_1(f)}{\left|1 - H_1(f)H_2(f)\right|^2}$$

$$(17)$$

This completes the exact decomposition of the CCH into four additive elements in the frequency domain. When the overall gain of the feedback loop is small, i.e., when $H_1(f)H_2(f) \to 0$, the last element and the denominator of Eq. (17) vanish, and the Fourier transform of the CCH simplifies to

$$\mathscr{F}(\text{CCH}_{12}) = \mathscr{F}(\text{CCH}_{12}^0) + \mathscr{F}(\text{ACH}_1^0)H_1(f) + \mathscr{F}(\text{ACH}_2^0)\overline{H_2(f)} \quad (18.1)$$

Which is represented in the time domain as

$$\text{CCH}_{12} = s_1^0 \star s_2^0 + \text{ACH}_1^0 * h_1 + \text{ACH}_2^0 * h_2(-\tau) \quad (18.2)$$

In Eq. (18.2), $\star$ denotes cross-correlation and $*$ denotes convolution. We have reached an important conclusion. When the overall gain of the feedback loop is small, the cross-correlation between two spike trains can be expressed as a sum of three elements: (1) the cross-correlation between the uncoupled spike trains; (2) the convolution of the auto-correlation of one uncoupled spike train with the impulse response of the coupling to the second train; and (3) the convolution of the auto-correlation of the second uncoupled train with time-reversed coupling to the first train.

The exact same derivation applies to inhibitory connections, the difference being that the plus signs in Eq. (11) and Eq. (12) are replaced by minus signs. In the absence of connectivity, Eq. (17) simplifies to $\text{CCH} = \text{CCH}_{12}^0$. Alternatively, in the presence of purely feedforward connectivity, e.g., when $\overline{H_2(f)} \to 0$, Eq. (17) simplifies to

$$\mathscr{F}(\text{CCH}_{12}) = \mathscr{F}(\text{CCH}_{12}^0) + \mathscr{F}(\text{ACH}_1^0)H_1(f) \quad (19.1)$$

Which is represented in the time domain as

$$\text{CCH}_{12} = s_1^0 \star s_2^0 + \text{ACH}_1^0 * h_1 \qquad (19.2)$$

In some systems, e.g., when neurons are strongly coupled[69,70], the approximation $H_1(f)H_2(f) \to 0$ cannot be made. In general, the approximation $H_1(f)H_2(f) \to 0$ can be made when the loop gain (rSTG$_{12}$ times rSTG$_{21}$) is lower than a minimal unidirectional STG, which can be detected with finite data. The detection of a connection depends on multiple variables including recording duration, firing rates, and the method employed. Therefore, an exact number for a minimal detectable unidirectional STG cannot be obtained. However, in the specific case of unidirectional connection with an impulse response spanning a single bin and Poisson firing pattern of both neurons, the minimal detectable unidirectional STG can be computed exactly using the tails predictor. Under these conditions, the CCH baseline is[20] $\lambda = F_1 F_2 TB$, where $F_1$ and $F_2$ are the firing rates of the pre- and postsynaptic neurons in spk/s, $T$ is the effective recording duration, and $B$ is the CCH bin width (both measured in seconds) For a given $\lambda$ and alpha (detection threshold) level, the inverse Poisson distribution can be used to obtain the minimal number of transmitted spikes required for detection. The minimal STG is then calculated as the minimal number of transmitted spikes, divided by $F_1 T$. For example, the minimal STG which can be detected for $F_1 = 1$ spk/s, $F_2 = 10$ spk/s, at $T = 50,000$ s using $B = 0.001$ s, and an alpha level of 0.001 is 0.00142.

**Algorithm for deconvolving ACHs from the CCH.** In general, we do not have access to the latent variables $s_1^0$ and $s_2^0$. In other words, we cannot determine which spikes in a train, e.g., $s_2$, are background spikes (e.g., $s_2^0$), and which are transmitted from $s_1$ (Eq. 11). Thus, in general, ACH$_1^0$ and ACH$_2^0$ are unknown and Eq. 18 cannot be evaluated. However, if the number of transmitted spikes is small, compared to background spikes, then the ACHs of the background trains can be estimated from the measured spike trains:

$$\text{ACH}_1^0(\tau) \approx \text{ACH}_1(\tau) = \int_0^\infty s_1(t)s_1(t+\tau)dt \qquad (20.1)$$

$$\text{ACH}_2^0(\tau) \approx \text{ACH}_2(\tau) = \int_0^\infty s_2(t)s_2(t+\tau)dt \qquad (20.2)$$

Where the lower integration bound is 0 rather than $-\infty$ due to the non-negativity of the time axis. The CCH is estimated directly from the spike trains as in Eq. (14.1), with the time axis modified in the same manner:

$$\text{CCH}_{12}(\tau) = \int_0^\infty s_1(t)s_2(t+\tau)dt \qquad (20.3)$$

Finally, we define

$$\widetilde{H}_1(f) = \frac{H_1(f)}{\mathscr{F}(\text{ACH}_2)} \qquad (21.1)$$

$$\widetilde{H}_2(f) = \frac{\overline{H_2(f)}}{\mathscr{F}(\text{ACH}_1)} \qquad (21.2)$$

Using these measurements (Eq. 20) and definitions (Eq. 21) in Eq. (18.1) and rearranging, we obtain

$$\widetilde{H}_1(f) + \widetilde{H}_2(f) + \frac{\mathscr{F}\left(\text{CCH}_{12}^0\right)}{\mathscr{F}(\text{ACH}_1)\mathscr{F}(\text{ACH}_2)} = \frac{\mathscr{F}\left(\text{CCH}_{12}\right)}{\mathscr{F}(\text{ACH}_1)\mathscr{F}(\text{ACH}_2)} \qquad (22)$$

Where the r.h.s. is derived from the observed spike trains. Taking the inverse Fourier transform and defining the "other inputs" element as

$$\text{OI} = \mathscr{F}^{-1}\left[\frac{\mathscr{F}\left(\text{CCH}_{12}^0\right)}{\mathscr{F}(\text{ACH}_1)\mathscr{F}(\text{ACH}_2)}\right] \qquad (23)$$

We obtain the basis of the deconvolution algorithm:

$$\widetilde{h}_1 + \widetilde{h}_2(-\tau) + \text{OI} = \mathscr{F}^{-1}\left[\frac{\mathscr{F}\left(\text{CCH}_{12}\right)}{\mathscr{F}(\text{ACH}_1)\mathscr{F}(\text{ACH}_2)}\right] \qquad (24.1)$$

Due to causality (Eq. 10), $h_1$ and $h_2(-\tau)$ reside on opposite sides of the zero lag. The r.h.s. of Eq. (24.1) is denoted the "deconvolved CCH" (dcCCH):

$$\text{dcCCH}_{12} = \mathscr{F}^{-1}\left[\frac{\mathscr{F}\left(\text{CCH}_{12}\right)}{\mathscr{F}(\text{ACH}_1)\mathscr{F}(\text{ACH}_2)}\right] \qquad (24.2)$$

When unidirectional deconvolution is applied, Eqs. (24.1–2) can be replaced by:

$$h_1 + \text{OI} = \mathscr{F}^{-1}\left[\frac{\mathscr{F}\left(\text{CCH}_{12}\right)}{\mathscr{F}(\text{ACH}_1)}\right] \qquad (24.3)$$

$$\text{dcCCH}_{12} = \mathscr{F}^{-1}\left[\frac{\mathscr{F}\left(\text{CCH}_{12}\right)}{\mathscr{F}(\text{ACH}_1)}\right] \qquad (24.4)$$

Once the dcCCH is determined numerically (see below), the "other inputs" element can be determined using timescale separation (a predictor; Eq. 6).

Following subtraction and rescaling (Eq. 7), the STCs and the eSTGs (Eq. 8) are evaluated.

The first step in the actual deconvolution algorithm is to compute the two count ACHs and the CCH from the spike trains. The second step is to scale the ACHs, such that each of the ACHs will have a sum of one. To scale the count ACH, the zero-lag bin is first set to zero. Then, the mean is subtracted, every bin in the zero-mean ACH is divided by the total number of spikes, and the zero-lag bin is set to complement the sum to one. Scaling ensures that deconvolution will be affected only by the shape of the ACHs, and not by the number of spikes in each train. If the count ACH is exactly flat, the normalization yields a delta-like histogram (one at zero-lag and zero everywhere else), and deconvolution does not modify the CCH. Third, the r.h.s. of Eq. (24) is evaluated.

The deconvolution algorithm as derived above (Eq. 20–24) is based on Eq. 18, but makes two key simplifying assumptions. First, that the shape of the ACH is not considerably distorted by the transmitted spikes (Eq. 20). Second, the impulse response is not considerably distorted by the divisive scaling factor inherent in the derivation (Eq. 21).

With respect to Eq. 20, using point process simulations, we found that even when the presynaptic neuron spikes at a relatively high rate (10 spk/s) with strong bursting activity (BR = 0.4), and the postsynaptic neuron spikes at a lower rate (3 spk/s), the estimates of connectivity are not distorted as long as the rSTG is below 0.25. Notably, strong connections and high firing rates are not characteristic of local cortical networks. However, different settings may appear in different systems, e.g., thalamocortical feedforward connections[28]. When connectivity is very strong and presynaptic firing rates are high, prior knowledge about connectivity can be used to employ unidirectional deconvolution (Eq. 24.3–4) instead of the "standard" deconvolution (Eqs. 24.1–2).

While ACHs effects are eliminated by the deconvolution algorithm, a divisive scaling factor (the denominator of Eq. 21) is introduced which can distort the recovered impulse responses. If connectivity is unidirectional and the postsynaptic train is Poisson, the scaled ACH$_2$ is a delta function, the divisive factor in Eq. (21.1) is unity, and $\widetilde{h}_1 = h_1$. Furthermore, if connectivity is unidirectional, then even if the presynaptic train is non-Poisson, the numerator of the r.h.s. of Eq. (21.2) is zero and thus $h_2 = 0$. When connectivity is bidirectional and one of the trains is non-Poisson, or when connectivity is unidirectional and the postsynaptic train is non-Poisson, the divisive factor may become consequential. Then, prior knowledge about connectivity can be used to employ unidirectional deconvolution (Eq. 24.3–4).

**Point process spike train simulations.** For investigating different STG estimation methods, we devised a point process simulation that generates two spike trains of a two-neuron network with precisely defined STGs. Each of the synthetic spike trains, $s_1(t)$ and $s_2(t)$, was initially generated by random sampling from a distinct rate function, $\lambda_1(t)$ and $\lambda_2(t)$, respectively, and then spikes were added or removed according to predetermined STCs. Thus, simulations consisted of three sequential steps: (1) generation of a rate function for each neuron; (2) stochastic sampling of spike trains, independently from each rate function; (3) modification of each spike train according to the other spike train and the corresponding STC. All simulations were generated over $N = T/\Delta t$ samples, where $\Delta t = 0.001$ s is the time step and $T$ is the duration of the simulation.

To generate a rate function for one neuron, we first set a desired mean firing rate $\lambda^d$ which was then modified based on the desired gamma order $\gamma$ and burst fraction $BR$ according to $\lambda^m = \lambda^d \cdot \gamma/(1 + BR)$. For instance, for generating a spike train with gamma order $\gamma = 2$ the firing rate is doubled, and for a spike train in which half of the spikes are first spikes in a two-spike burst the firing rate is halved. For generating a fixed rate function, we set $\lambda(t) = \lambda^m \ \forall t \in [1\ N]$, separately for each neuron. To create co-modulated spike trains, we generated correlated rate functions for the two neurons. For that, the rate function of each neuron was time dependent, defined as: $\lambda(t) = \lambda^m + \lambda^m \cdot \lambda_c(t)$. The co-modulation rate function $\lambda_c$ is a clipped pink noise signal, generated by filtering Gaussian white noise sampled at $\Delta t$ intervals, $\eta(t) \sim G(0,\sigma_c)$, with a decaying exponential $e^{-1/\tau_c}$, where $\tau_c = 20$ ms. The pink noise was then clipped to the $[-1\ 1]$ range, yielding the zero-mean $\lambda_c(t)$. The same co-modulation function was used for generating the two rate functions, $\lambda_1(t)$ and $\lambda_2(t)$.

In the second step, each spike train was generated by stochastic sampling from the corresponding rate function. For each sample, we determined whether a spike did or did not occur, according to random sampling from a Binomial distribution with parameters $B(n,p) = (1,\lambda(t)\Delta t)$. When $\lambda(t)$ is time-varying or constant, the resulting spike trains correspond to non-homogenous or homogenous Poisson processes, respectively. After generating a spike train, three modifications were made. First, spikes were removed according to the gamma order $\gamma$: to realize an $n$th-order gamma process, every $n$th spike was retained, and all other spikes were decimated. Retaining all spikes corresponds to a first-order gamma (i.e., Poisson) process. Second, burst spiking activity was added by inserting spikes based on the burst fraction ($BR$) parameter. For a non-zero $BR$, each spike was defined to be a "first in burst" randomly with probability $BR_1$. The precise timing of the "second in burst" spike was determined by random (uniform) sampling from a 5-element symmetric triangular window that lagged the first spike in the burst by two samples. A "third in burst" spike was added with probability $BR_2$; then, the overall $BR$ is $BR_1 + BR_1 \cdot BR_2$, and the timing of the third spike was drawn from a three-sample lagged window. In actual simulations, $BR_1$ was varied in the [0 0.4] range,

whereas $BR_2 = 0.4$ was held constant. Third, ARP considerations were imposed by removing all spikes that occurred shortly (ARP = 2 ms) after another spike.

In the third step, connectivity was added between the two spike trains. The same STC waveform was used in all simulations: a 5-element asymmetric triangular window that lagged the presynaptic spike by one sample. However, the STC integral (the STG) was varied in sign and value, mimicking excitatory and inhibitory connectivity of various gains. For each spike of the presynaptic neuron, the occurrence and timing of the added (or decimated, for inhibitory connections) spike was determined by random (uniform) sampling from the STC. For excitatory transmission, this procedure can generate more than one postsynaptic spike, as required since the STG is not limited to the [0 1] range. Connectivity was implemented independently in every direction. After applying connectivity between the two trains, ARP considerations were imposed once more.

For creating noisy point process simulations (Fig. 4), we simulated 1250 spike train pairs. In 60% of the pairs, co-modulation dynamics were added. For the simulations with co-modulation, the co-modulation parameter $\sigma_c$ was drawn randomly from a uniform distribution between 1 and 15 spk/s, with a fixed co-modulation time constant ($\tau_c = 20$ ms). The firing rates of the pre- and postsynaptic neurons were 2 and 8 spk/s, respectively. The burst fraction of the presynaptic neuron was varied uniformly between 0 and 0.4. Recording duration was set randomly between 90 min and 5 h. Of the 1250 simulated pairs, 500 pairs were simulated with monosynaptic excitatory connections; 500 pairs were simulated with monosynaptic inhibitory connections; and the other 250 pairs were simulated without connectivity. Connectivity strengths (rSTG) were drawn randomly from a log-normal distribution[43–45] with {mean, SD} of {0.019, 0.01} for the excitatory connections, and {−0.014, 0.007} for the inhibitory connections.

**Conductance-based model of CA1 network.** To model synaptic connectivity within a network of neurons with ACHs and CCHs resembling those observed in real neuronal data (Fig. 6), we generated a network of conductance-based E- and I-cells. The E-cell model, derived from the "simple model" of Izhikevich[71], included dynamics on the membrane potential ($V$), and a slower, phenomenological recovery variable ($u$). In addition, the model included synaptic input and noise. The model equations are:

$$\begin{cases} C\frac{dV}{dt} = I_{in}^e(t) + V_k\left(V - E_L^e\right)\left(V - V_t\right) - u - I_{synaptic} + g_N\eta(t) \\ \frac{du}{dt} = U_a\left(U_b\left(V - E_L^e\right) - u\right) \\ \text{if } V > V_{peak} \text{ then } V \leftarrow V_{reset}; u = u + U_{step} \end{cases} \quad (25)$$

Membrane potential variability, which may stem from many unknown sources (e.g., unrecorded neurons), was modeled by an additive noise term, generated by random sampling from a zero-mean Gaussian distribution $\eta(t) \sim N(0,\sigma)$ independently for every cell. In this model, spike waveforms were modeled explicitly by the quadratic in Eq. 25. Whenever a spike occurred (i.e., $V_{peak}$ was crossed), the membrane potential $V$ was reset to $V_{reset}$, and the recovery variable $u$ was incremented by $U_{step}$. By modifying these two parameters, a given E-cell model was tuned to fire predominantly single spikes, spike bursts, or mixes thereof (Fig. 5a). The specific parameters values used for the E-cell model are detailed in Table 1.

For the I-cells, we used the Wang-Buzsáki model[72], describing the dynamics of the membrane potential ($V$), sodium inactivation ($h$), and delayed-rectifier potassium ($n$). The full model also included synaptic currents and noise, and reads

$$\begin{cases} C\frac{dV}{dt} = I_{in}^i(t) - g_L\left(V - E_L^i\right) - g_{Na}hm_\infty(V)^3\left(V - E_{Na}\right) - g_K n^4\left(V - E_K\right) - I_{synaptic} + \eta(t) + g_N\eta(t) \\ \frac{dh}{dt} = \frac{h_\infty(V)-h}{\tau_h(V)} \\ \frac{dn}{dt} = \frac{n_\infty(V)-n}{\tau_n(V)} \end{cases}$$

$$(26.1)$$

**Table 1 Parameters used for modeling E-cells in the conductance-based networks.**

| Parameter | Value | Units |
|---|---|---|
| $C$ | 1 | μF/cm² |
| $V_k$ | 0.01 | mS/(cm²·mV) |
| $E_L^e$ | −60 | mV |
| $V_t$ | −45 | mV |
| $g_N$ | 1 | mS/cm² |
| $U_a$ | 0.02 | mS/cm² |
| $U_b$ | 0.01 | mS/cm² |
| $V_{peak}$ | 40 | mV |
| $V_{reset}$ | [−42 −35] | mS/cm² |
| $U_{step}$ | [1.12 1.225] | mV·mS/cm² |
| $\sigma^e$ | 2 | mV |
| $I_{in}^e$ | 0 | μA/cm² |
| $N_e$ | 80 | |

**Table 2 Parameters used for modeling I-cells in the conductance-based networks.**

| Parameter | Value | Units |
|---|---|---|
| $C$ | 1 | μF/cm² |
| $g_L^i$ | 0.1 | mS/cm² |
| $E_L^i$ | −65 | mV |
| $g_{Na}^i$ | 35 | mS/cm² |
| $E_{Na}^i$ | 55 | mV |
| $g_K^i$ | 9 | mS/cm² |
| $E_K^i$ | −90 | mV |
| $g_N^i$ | 1 | mS/cm² |
| $\sigma^i$ | 2 | mV |
| $I_{in}^i$ | −0.75 | μA/cm² |
| $N_i$ | 20 | |

The gating variables for the I-cell ($x = h,m,n$) had voltage-dependent time constants ($\tau_x$) and steady-state values ($x_\infty$) as follows:

$$h_\infty(V) = \frac{0.07e^{\frac{-(V+58)}{20}}}{0.07e^{\frac{-(V+58)}{20}} + \frac{1}{1+e^{\frac{-(V+28)}{10}}}}, \quad \tau_h(V) = \frac{0.2}{0.07e^{\frac{-(V+58)}{20}} + \frac{1}{1+e^{\frac{-(V+28)}{10}}}} \quad (26.2)$$

$$m_\infty(V) = \frac{\frac{0.2(V+35)}{1-e^{\frac{-(V+35)}{10}}}}{\frac{0.2(V+35)}{1-e^{\frac{-(V+35)}{10}}} + 4e^{\frac{-(V+60)}{18}}} \quad (26.3)$$

$$n_\infty(V) = \frac{\frac{0.01(V+34)}{1-e^{\frac{-(V+34)}{10}}}}{\frac{0.01(V+34)}{1-e^{\frac{-(V+34)}{10}}} + 0.125e^{\frac{-(V+44)}{80}}}, \quad \tau_n(V) = \frac{0.2}{\frac{0.01(V+34)}{1-e^{\frac{-(V+34)}{10}}} + 0.125e^{\frac{-(V+44)}{80}}} \quad (26.4)$$

Other parameters values used for the I-cell model are detailed in Table 2. Synaptic connections were modeled as in ref. [73]. For the $e$'th E-cell, the total synaptic current was

$$I_{synaptic,e} = \sum_{j=1}^{N_e} g_{ej}S_{ej}\left(V_e - E_{se,j}\right) + \sum_{k=1}^{N_i} g_{ek}S_{ek}\left(V_e - E_{si,k}\right) \quad (27.1)$$

Where $N_e$ ($N_i$) is the number of E-cells (I-cells). The notation $g_{ej}$ indicates the maximal synaptic conductance from presynaptic E-cell $j$ to postsynaptic E-cell $e$. All excitatory-to-excitatory (E-to-E) synapses had the same reversal potential, regardless of the presynaptic neuron ($E_{se,j} = E_{se}, \forall j$), but the maximal excitatory conductance value $g_{ej}$ could vary between presynaptic neurons. In practice, we set all $g_{ej} = g_{ee} = 0$. All inhibitory-to-excitatory (I-to-E) synapses had the same reversal potential, regardless of the presynaptic neuron ($E_{si,k} = E_{si}, \forall k$). The maximal inhibitory conductance $g_{ek}$ was either fixed for all I-cells ($g_{ek} = g_{ei}$; Fig. 5d) or varied between neurons (Fig. 5i). All synaptic activation variables corresponding to the same presynaptic neuron had the same dynamics, regardless of the postsynaptic neuron ($S_{ej} = S_j$, $S_{ek} = S_k$, $\forall e$). For the $i$'th I-cell, the total synaptic current was modeled by

$$I_{synaptic,i} = \sum_{j=1}^{N_e} g_{ij}S_{ij}\left(V_i - E_{se,j}\right) + \sum_{k=1}^{N_i} g_{ik}S_{ik}\left(V_i - E_{si,k}\right) \quad (27.2)$$

All excitatory-to-inhibitory (E-to-I) synapses had the same reversal potential ($E_{se,j} = E_{se}, \forall j$), but the maximal conductance values, $g_{ip}$, could be varied between E-cells. All inhibitory-to-inhibitory (I-to-I) synapses had the same reversal potential ($E_{si,k} = E_{si}, \forall k$), but the maximal conductance values $g_{ik}$ could be varied between I-cells. All synaptic activation variables corresponding to the same presynaptic neuron had the same dynamics ($S_{ij} = S_j$, $S_{ik} = S_k$, $\forall i$).

For an excitatory/inhibitory presynaptic neuron, the dynamics of the corresponding synaptic variable ($S_e/S_i$) depended on the presynaptic membrane potential ($V_e/V_i$) and the synaptic rise and decay time constants, following:

$$\frac{dS_e}{dt} = H\left(V_e\right)\frac{\left(1 - S_e\right)}{\tau_r^e} - \frac{S_e}{\tau_d^e} \quad (28.1)$$

$$\frac{dS_i}{dt} = H\left(V_i\right)\frac{\left(1 - S_i\right)}{\tau_r^i} - \frac{S_i}{\tau_d^i} \quad (28.2)$$

$$H(V) = \left(1 + \tanh\left(V/4\right)\right)/2 \quad (28.3)$$

All synaptic parameters values used are detailed in Table 3. Numerical integration was done using the explicit second-order Runge-Kutta endpoint (modified Euler) method with integration time step of $\Delta t = 0.025$ ms and simulation duration of $T_{sim} = 240$ min.

**Animals and ethics.** Three freely-moving male C57BL/6J mice were used in this study (Table 4). At the time of implantation, mice aged 14–16 weeks and weighed 26–30 g. After implantation, mice were single-housed to prevent damage to the implanted apparatus. All animal handling procedures were in accordance with Directive 2010/63/EU of the European Parliament, complied with Israeli Animal Welfare Law (1994), and approved by the Tel Aviv University Institutional Animal Care and Use Committee (IACUC #01-16-051).

**Probes and surgery.** Each animal was implanted with a multi-shank silicon probe (Diagnostic Biochips) attached to a movable micro-drive. The probes used were Stark64 (two mice) and Dual-sided64 (one mouse). The Stark64 probe consists of six shanks, spaced horizontally 200 μm apart, with each shank consisting of 10–11 recording sites, spaced vertically 15 μm apart. The Dual-sided64 probe consists of two dual-sided shanks, spaced horizontally 250 μm apart, with each shank consisting of 16 channels on each side (front and back), spaced vertically 20 μm apart. In all mice, probes were implanted in the neocortex above the hippocampus (PA/LM, 1.6/1.1 mm) under isoflurane (1%) anesthesia[38,74]. After every recording session, the probe was translated vertically downward by up to 70 μm. The recorded data included only recordings from the CA1 pyramidal cell layer, recognized by the appearance of multiple high-amplitude spiking units and iso-potential spontaneous ripple events.

**Recording procedures.** Neuronal activity was recorded in 4.4-h sessions (median of 27 sessions; range, 3–12 h). Every session started with a baseline neural recording of at least 15 min, while the animal was in the home cage or in a 0.8 m diameter open field. After the baseline recordings, the animal ran on a linear track, received optogenetic stimuli, or both, for a period of at least 60 min. Sessions ended with another baseline period of at least 30 min. Only data recorded during spontaneous activity, in the lack of any optogenetic stimuli, were used in this work.

**Spike detection and sorting.** Neural activity was filtered, amplified, multiplexed, digitized on the headstage (0.1–7500 Hz, x192; 16 bits, 20 kHz; RHD2132 or RHD2164, Intan Technologies), and recorded by an RHD2000 evaluation board (Intan Technologies). Offline, spikes were detected and sorted into single units automatically using either KlustaKwik3[75,76] or KiloSort2[77]. Automatic spike sorting was followed by manual adjustment of the clusters. Only well-isolated units were used for further analyses (amplitude >40 μV; L-ratio <0.05[78]; ISI index <0.2[79]). Units were classified into putative pyramidal cells (PYR) or parvalbumin-immunoreactive [PV]-like inhibitory interneurons (INT) using a Gaussian mixture model[29].

**Quantifying burst spiking activity.** For quantifying burst spiking activity of real spike trains, we used a "burst index"[80]. The precise definition employed was

burst index $= \frac{\text{head}-\text{tail}}{\text{head}+\text{tail}}$, where head is the sum of all ACH counts in the $2 < \tau \le 10$ ms range, and tail is the sum of all ACH counts in the $35 < \tau \le 50$ ms range.

**Selection of a subset of data.** For evaluating the contribution of deconvolution to the analysis of real neuronal data, we used spike trains of units recorded from CA1 of three freely-moving mice during spontaneous activity. We denote PYR-to-INT pairs for which both eSTG$_{median}$ and eSTG$_{dc-median}$ were positive as the "All pairs" group (7991/9030 simultaneously-recorded PYR-INT pairs; Fig. 6d, blue dots). For eSTG comparisons, we focused on a subset of pairs (the "Analysis pairs" group; 1274/7991 pairs; Fig. 6d, pink dots) composed of pairs which met four criteria. (1) The trigger unit (PYR) exhibited a non-negative burst index (Fig. 6c). (2) An excitatory connection was detected by the median filter with and without deconvolution. (3) The ratio between eSTG$_{median}$ and eSTG$_{dc-median}$ was between 0.5 and 2 (28 pairs were excluded due to this criterion). (4) The total number of counts in the count CCH (at the $-30 < \tau \le 30$ range) was above 400 (5 pairs were excluded due to this criterion).

**Detection analysis.** For quantifying detection performance of each method, we first computed false-positive (FP) and false-negative (FN) rates for each type of connection. For excitatory connections, the FP rate was defined as the number of times in which the method detected an excitatory connection when no connection was simulated, divided by the total number of tested connections ($N_{conn}$). The excitatory FN rate was defined as the number of times in which an excitatory connection was simulated but was not detected by the method, divided by $N_{conn}$. For inhibitory connections, the FP rate was defined as the number of times in which the method detected an inhibitory connection when no connection was simulated, divided by $N_{conn}$. The inhibitory FN rate was defined as the number of times in which an inhibitory connection was simulated but was not detected by the method, divided by $N_{conn}$. Next, we computed the $f_1$ score, defined as:

$$f_1 = \frac{\text{TP}}{\text{TP} + 0.5(\text{FP} + \text{FN})} \qquad (29)$$

Where TP is the true positive rate, which is defined as the number of times in which a simulated connection (excitatory or inhibitory) was correctly detected by the method, divided by $N_{conn}$. The FP and FN used in Eq. (29) are the sum of the excitation and inhibition FP and FN rates: FP = FP$_{exc}$ + FP$_{inh}$ and FN = FN$_{exc}$ + FN$_{inh}$, respectively.

To determine whether the $f_1$ score of one group (group$_1$) is consistently larger than the $f_1$ score of another group (group$_2$), we devised a resampling with replacement (bootstrap) test. For each resampled dataset, we first computed the $f_1$ score for each group, and then compared the two $f_1$ scores. We repeated the process 3500 times. We then counted the fraction of times for which group$_1$ $f_1$ score was smaller than group$_2$ $f_1$ score. Based on the resampled scores, we computed the mean and SEM.

**Slope comparison.** To determine whether the linear slope of one group (group$_1$) is consistently larger than the slope of another group (group$_2$) we devised another, more conservative bootstrap test. For each resampled dataset, we fitted a slope for each group. We repeated the process 300 times, resulting in 300 different slopes for each group and 90,000 slope pairs. We then counted the fraction of pairs for which group$_1$ slope was smaller than group$_2$ slope. Pairing slopes from different resampling iterations provides a more conservative test than only pairing slopes from the same iteration.

**Statistics and reproducibility.** In all statistical tests used in this study, a significance threshold of $\alpha = 0.05$ was used. Two exceptions were the threshold used for determining whether two units exhibit monosynaptic connectivity, and the classifier used to determine whether a unit is a PYR or an INT ($\alpha = 0.001$). For Fig. 5m, reproducibility was carried out across networks (Fig. 5m); for Fig. 6g, reproducibility was carried out across animals (Table 4). All descriptive statistics ($n$, mean, median, range, IQR, SEM) can be found in the results and figure legends. To examine whether a group median is larger or smaller than an expected value, we used Wilcoxon's signed-rank test (one-tailed). Differences between two group medians were tested with Mann–Whitney's $U$-

**Table 3 Synaptic parameters used in simulating conductance-based networks.**

| Parameter | Value | Units | Notes |
|---|---|---|---|
| $\tau_r^e$ | 0.1 | ms | AMPA |
| $\tau_d^e$ | 3 | ms | AMPA |
| $E_e$ | 0 | mV | AMPA |
| $\tau_r^i$ | 0.3 | ms | GABA$_A$ |
| $\tau_d^i$ | 9 | ms | GABA$_A$ |
| $E_i$ | −80 | mV | GABA$_A$ |
| $g_{ie}$ | 0.055 | mS/cm$^2$ | Fig. 5i: mean: 0.055, SD: 0.02 |
| $g_{ee}$ | 0 | mS/cm$^2$ | |
| $g_{ei}$ | 0.25 | mS/cm$^2$ | Fig. 5i: mean: 0.22, SD: 0.08 |
| $g_{ii}$ | 0.25 | mS/cm$^2$ | Fig. 5i: mean: 0.22, SD: 0.04 |

**Table 4 List of experimental animals.**

| Animal ID | Sex | Age[a] [week] | Weight[a] [g] | Probe | Sessions | All PYR[b] | All INT[b] | Analysis PYR[c] | Analysis INT[c] | Analysis pairs[c] | Rank CC[d] | P-value[e] |
|---|---|---|---|---|---|---|---|---|---|---|---|---|
| mDS1 | Male | 14 | 25.7 | Dual-sided64 | 9 | 322 | 50 | 170 | 37 | 427 | −0.19 | 0.0003 |
| mP30 | Male | 14 | 28.6 | Stark64 | 8 | 277 | 57 | 184 | 39 | 387 | −0.3 | 0.0003 |
| mP31 | Male | 16 | 30 | Stark64 | 10 | 442 | 108 | 257 | 76 | 458 | −0.09 | 0.0776 |
| Summary | | | | | 27 | 1041 | 215 | 611 | 152 | 1272 | −0.35 | 0.0003 |

[a]At the time of implantation.
[b]"All PYR" and "All INT" refer to the total numbers of well-isolated units recorded during the listed sessions.
[c]"Analysis PYR", "Analysis INT", and "Analysis pairs" refer to the subsets used for actual analyses (cf. Fig. 6b–d).
[d]"Rank CC" refers to Spearman's rank correction coefficient between eSTG ratios and burst indices.
[e]"P-value" refers to permutation test comparing the rank CC to a zero null.

test for unpaired samples (two-tailed), or with Wilcoxon's signed-rank paired test for paired samples (two-tailed). Association between parameters was estimated with Spearman's rank correlation and tested using a permutation test.

**Reporting summary**. Further information on research design is available in the Nature Research Reporting Summary linked to this article.

## Data availability

The source data behind the graphs in the paper are available in Supplementary Data 1. The data used in this study are available from the corresponding author upon reasonable request.

## Code availability

Code for implementing deconvolution and median filtering, along with example spike data, are hosted publicly on GitHub, accessible via https://github.com/EranStarkLab/CCH-deconvolution.

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

## Acknowledgements

This work was supported by ERC #679253, ISF #638/16, Rosetrees #A1576, and CIHR-IDRC-ISF #2558/18. We thank Ortal Amber-Vitos, Refaela Atsmon, Roni Gattengo, Tom Manovitz, Shir Sivroni, and Alexander Tarnavsky-Eitan for their constructive comments.

## Author contributions

L.S., A.L., S.S., and E.S. built optoelectronic devices; L.S. and E.S. implanted animals, developed analyses, and wrote the paper; L.S., A.L., and H.S. collected data; L.S. analyzed data and prepared figures; E.S. developed the mathematical framework and the deconvolution algorithm, conceived, and supervised the project.

## Competing interests

The authors declare no competing interests.

## Additional information

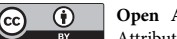

