## [Peer Review File · Communications Biology]

Reviewers' comments:

Reviewer #1 (Remarks to the Author):

This manuscript developed a new method to quantify the spike transmission probability (or spike transmission gain, STG) more accurately from spike trains of two neurons using deconvolution. This method is based on the authors' assumption that the cross-correlation histogram of spike trains of two neurons can be regarded as a sum of auto-correlation histogram (Neuron 1) * spike transmission curve (spike transmission probability from Neuron 1 to 2 as a function of time-lag from Neuron 1 spiking), auto-correlation histogram (Neuron 2) * spike transmission curve (spike transmission probability from Neuron 2 to 1 as a function of time-lag from Neuron 2 spiking), and cross-correlation histogram of uncoupled spike trains (* is convolution); therefore the cross-correlation histogram can be decomposed to yield the spike transmission curve. Using simulated spike trains and conductance-based leaky integrate and fire model neurons, the authors showed that spike transmission probability (and post-synaptic potential) could be better estimated with deconvolution than without deconvolution. In addition, they showed that deconvolution is effective especially when spike trains contain burst spikes. Further, the authors show that the spike transmission probabilities estimated by the deconvolution-based method are higher in the presence of burst spikes in the real neuronal data, which is similar to their result of artificially generated spike trains. I think the methods and experiments used in this manuscript are sound.

Estimating spike transmission probability between spike trains is widely performed by many researchers and potentially a powerful method to determine the effective connectivity between neurons. Thanks to recent progress in large-scale recording methods, estimating spike transmission probability among many neurons can potentially reveal our brain's functional architecture. I'm very enthusiastic about this manuscript because it improves the detection and estimation of the spike transmission probability, thus it is helpful for many researchers and can significantly contribute to the research field.

The manuscript is well written. I have only one minor issue for this paper.

1. Line 793: "Connectivity strengths were drawn randomly from a log-normal distribution." The authors should validate the choice of distribution. The magnitude of EPSP is often reported to follow log-normal distribution in the real data in the literature, but how about the size of IPSP?

Reviewer #2 (Remarks to the Author):

This manuscript points to possible limitations of existing methods in quantifying the transmission gain between neurons, and overcome those. Such gains assume importance in studying neural connectivity. In this regard, reported methods based on cross-correlation histograms (CCH) could be restrictive because neuronal responses may be correlated even without a direct connection (under a common influence). Existing time-scale separation techniques identify direct connections as corresponding to fast transients in the CCH, but tend to be confounded when one neuron exhibits burst spiking. This paper attempts at overcoming this.

In particular, they present a decomposition of CCH into three components (assuming bidirectional connection): the first one becomes the third one when the presynaptic and postsynaptic neurons are exchanged; the second term is symmetric (and relevant for bursty behavior). These components are consistent with basic concept of signals and linear systems. Their framework/model allows them to mitigate artefacts arising from bursty spiking using deconvolution. I believe that the premise and the rationale of this manuscript are generally sound. The authors also demonstrate the method on simulated data as well as real-life data collected from CA1 area of the hippocampus of freely moving mice.

In general, the paper is well planned, organized and presented. It should be fit for publication, if certain concerns are addressed.

My main concern is the following. Isn't it possible for two directly connected neurons to exhibit bursty behavior? If yes, how do you cater to this case? If no, why is this an unlikely scenario?

The above concern arises because burst spiking plays an important role in high level functions. So, bursts as artifacts... is that always justified? In this connection, the following reference appears to indicate that burst spiking in presynaptic neurons causes spiking in postsynaptic neurons.

Zeldenrust, Fleur, Wytse J. Wadman, and Bernhard Englitz. "Neural coding with bursts—current state and future perspectives." *Frontiers in computational neuroscience* 12 (2018): 48.

<https://www.frontiersin.org/articles/10.3389/fncom.2018.00048/full>

Or, is this a characteristic of neuronal responses from the CA1 area in the hippocampus only? Please clarify.

My other comments are minor.

1. While CCH is defined, ACH is not.
2. Figures should be presented in better quality. Font size must be increased (difficult to read at present).
3. Why a bin size of 1 ms? Please clarify, possibly in relation to the following reference.

Caro-Martín, Carmen Rocío, José M. Delgado-García, Agnes Gruart, and R. Sánchez-Campusano. "Spike sorting based on shape, phase, and distribution features, and K-TOPS clustering with validity and error indices." *Scientific reports* 8, no. 1 (2018): 1-28.
<https://www.nature.com/articles/s41598-018-35491-4>

4. How did you choose the temporal region (time interval) of interest as [0 5] ms while finding STC?

5. In Figure 5B(b), we see inhibitory conductance $G_{ei} = 0.25$. It is 0.025 in the description. Please rationalize.

Reviewer #3 (Remarks to the Author):

This paper introduces a new pre-processing approach (called 'deconvolution') to improve connectivity estimates between pairs of neurons based on their cross-correlation histogram (CCH). The authors claim that deconvolution reduces the artifacts introduced by the two individual auto-correlation histograms (ACHs) on the CCH, thus facilitating more accurate connectivity estimates. They convincingly validate these claims through comprehensive simulations and an application on real data, in particular when the spike trains exhibit bursting activities. The manuscript is well-written and the overall organization of the content is very good. Since more accurate methods to derive functional connectivity from emerging neural data acquisition techniques such as multi-electrode arrays are becoming key to understanding how the brain works, this could indeed be a consequential scientific contribution to the neuroscience society. However, while the results are plausible, there are several major concerns about the mathematical framework and the scope of the proposed method, which are required to be addressed.

1. In Eq. (7.2), the authors claim that the impulse response from neuron 2 to 1, h_2 is non-zero for negative time lags, and that is the 'causal direction' from 2 to 1. Since this is not a very standard definition of causality, it would be helpful if this could be explained in more detail.
2. In Eq. (12), the denominator should be corrected as $|1-H_1(f)H_2(f)|^2$, since it is a product of

a complex number and its conjugate. This error has then propagated to Eq. (14) as well.

3. The authors claim that the last element and the denominator of Eq. (14) vanished if the feedback loop: $H_1(f)H_2(f)$ is small. For both of those terms to vanish, both $H_1(f)H_2(f)$ and $H_1(f)\bar{H_2(f)}$ need to go to zero. The only way that both of these quantities seem to go to zero is if at least one of $H_1(f)$ or $H_2(f)$ independently goes to zero. It would be better if the authors can elaborate the settings that these assumptions hold and does not hold in practice, for the benefit of the future users of these methods.

4. In Eq. (15.1) and (16.1), the second term on the RHS should be corrected as $F(ACH_1^0)\bar{H_1(f)}$. This error then propagates to all subsequent steps. For example, in Eq. (15.2) and (16.2), the second term on the RHS should be corrected as $ACH_1^0 h_1$. Further, the numerator of the RHS of Eq. (18.1) should also be corrected as $\bar{H_1(f)}$.

5. Then, since $\tilde{h_1}$ is a time reversed version of h_1 (because of the conjugation mentioned above), $\tilde{h_1}$ and $\tilde{h_2}$ are going to lie in the same side of the zero lag (since the authors originally claimed that h_1 and h_2 reside on the opposite sides). This makes the statement in Page 27: line 721 to be irrelevant. It is also noteworthy that had the authors originally claimed that both h_1 and h_2 are non-zero only for positive lags (which is the standard notion of causality), $\tilde{h_1}$ and $\tilde{h_2}$ would have actually laid on the opposite sides of the zero lag.

6. A fundamental assumption in this derivation is that if the number of transmitted spikes is small, then $ACH_1^0 \sim ACH_1$ and $ACH_2^0 \sim ACH_2$. For potential reproducibility and future usage, it is important to know what settings are required to meet these assumption. Thus, it would be useful if the authors can give an idea about the practical settings where these assumptions hold true, and if possible demonstrate through simulations cases where their assumptions are no longer valid.

7. While it is true that the multiplicative scaling of the filters $H_1(f)$ and $H_2(f)$ by the ACHs is eliminated by the deconvolution proposed in Eq. (21.2), it is also important to note that this is achieved at the cost of introducing a new divisive scaling factor. After this deconvolution, the recovered variables are $\tilde{h_1}$ and $\tilde{h_2}$, which are again scaled versions of the desired quantities h_1 and h_2 . It would be great if the authors could justify why this newly added scaling factor is not consequential. This is very important because the authors claim that the 'deconvolution results in a CCH which is free from ACH artifacts' (e.g. Page 10: line 231,234), which does not seem to be completely true according to their derivation.

8. While the authors provide extensive simulations showing that deconvolution improves the connectivity estimates when the spike trains exhibit bursting activities, they also claim that the deconvolution can remove 'second order spike train statistics from the CCH' (Page 2: line 61, Page 10: line 228 and Page 20: line 484). It would be helpful if they can provide insights on settings other than bursting where their method yields better connectivity estimates.

Following are some other minor comments to improve the readability of the manuscript.

1. Page 7: it would be helpful if the operators $*$ and \star are explicitly defined here as well. Also, it would be better if the abbreviation ACH is introduced in the text before the equation in line 159.
2. Page 12: Figure 3-A-a, please label the y axes and the individual subplots.
3. Page 12: Figures 3-A-d, 3-C-c and 3-F-c, it is not very clear how/whether the orange insets are related to the black and grey curves.
4. Page 17: Figure 5-C-b, the y-axis label is incorrect, it should be corrected as eSTG.
5. Page 22: line 536 & line 538, it would be better to replace all instances of S with S(t) to avoid confusion.
6. Page 22: line 544, please define the abbreviation ARP here, where it is first being used.

Deconvolution improves the detection and quantification of spike transmission gain from spike trains

We would like to thank the Reviewers for their insightful and constructive criticism. We prepared our revised manuscript in light of their comments.

Reviewer #1

Comment #1

Line 793: “Connectivity strengths were drawn randomly from a log-normal distribution.” The authors should validate the choice of distribution. The magnitude of EPSP is often reported to follow log-normal distribution in the real data in the literature, but how about the size of IPSP?

Addressing comment #1

Indeed, we used a log-normal distribution to draw inhibitory rSTGs in the point process simulations (**Fig. 4**), and to draw inhibitory conductance (Gei) values in the conductance-based simulations (**Fig. 5**). We validate the choice of distribution by reviewing relevant scientific literature. In three studies, CA1 pyramidal cells (PYR) were reported to show log-normal or skewed distributions of IPSCs (Edwards et al., 1990; Zaninetti and Raggenbass, 2000; Maniezzi et al., 2019). In two other studies, neocortical PYR were reported to show a skewed distribution of IPSCs (Perrais and Ropert, 1999; Nusser et al., 2001). Other studies have shown skewed distributions of IPSCs for dentate gyrus granule cells (Soltesz et al., 1995) and for cerebellar stellate cells (Nusser et al., 1997). Yet another study (Hoffmann et al., 2015) showed IPSP distributions for neocortical, and although the type of distribution was not stated it appeared skewed.

Following this comment, we now cite Buzsáki and Mizuseki (2014), Maniezzi et al. (2019), and Zaninetti and Raggenbass (2000) when stating the choice of distribution (**Results, pg. 15, line 316; Methods, pg. 32, line 869; References, pg. 42, lines 1128-1134**).

References:

Edwards, F. A., Konnerth, A. & Sakmann, B. Quantal analysis of inhibitory synaptic transmission in the dentate gyrus of rat hippocampal slices: a patch-clamp study. *The Journal of Physiology* 430, 213–249 (1990).

Hoffmann, J. H. O. et al. Synaptic Conductance Estimates of the Connection Between Local Inhibitor Interneurons and Pyramidal Neurons in Layer 2/3 of a Cortical Column. *Cerebral Cortex* 25, 4415–4429 (2015).

Maniezzi, C., Talpo, F., Spaiardi, P., Toselli, M. & Biella, G. Oxytocin Increases Phasic and Tonic GABAergic Transmission in CA1 Region of Mouse Hippocampus. *Frontiers in Cellular Neuroscience* 13, 178 (2019).

Nusser, Z., Cull-Candy, S. & Farrant, M. Differences in Synaptic GABAA Receptor Number Underlie Variation in GABA Mini Amplitude. *Neuron* 19, 697–709 (1997).

Nusser, Z., Naylor, D. & Mody, I. Synapse-Specific Contribution of the Variation of Transmitter Concentration to the Decay of Inhibitory Postsynaptic Currents. *Biophysical Journal* 80, 1251–1261 (2001).

Perrais, D. & Ropert, N. Effect of Zolpidem on Miniature IPSCs and Occupancy of Postsynaptic GABAA Receptors in Central Synapses. *Journal of Neuroscience* 19, 578–588 (1999).

Soltesz, I., Smetters, D. K. & Mody, I. Tonic inhibition originates from synapses close to the soma. *Neuron* 14, 1273–1283 (1995).

Zaninetti, M. & Raggenbass, M. Oxytocin receptor agonists enhance inhibitory synaptic transmission in the rat hippocampus by activating interneurons in stratum pyramidale. *European Journal of Neuroscience* 12, 3975–3984 (2000).

Reviewer #2**Comment #1**

My main concern is the following. Isn't it possible for two directly connected neurons to exhibit bursty behavior? If yes, how do you cater to this case? If no, why is this an unlikely scenario?

Addressing comment #1

Indeed, the evaluation of the deconvolution method was done using neuronal pairs in which only one neuron exhibits burst spiking activity. This is typical for PYR to INT pairs in hippocampal CA1 region. Yet as the Reviewer indicated, two directly-connected neurons may both exhibit burst spiking activity. This may be observed in the neocortex and in hippocampal CA3, among other regions.

To cater for the case of two directly-connected bursting neurons, we simulated two spike trains coupled by an excitatory monosynaptic connection ($rSTG=0.04$; 833 min; **Fig. R1Ac**). Both spike trains were initially simulated as Poisson processes modified by refractoriness ($\lambda=2$ spk/s; ARP=2 ms) and exhibited bursting activity (**Fig. R1Aa-b**). The burst fraction (BR) of the presynaptic neuron was set to 0.4, and the BR of the postsynaptic neuron was set to 0.1. Consistent with the construction, the CCH between the coupled spike trains exhibited a peak within the causal ROI, as well as side lobes due to the burst spiking activity of the two neurons (**Fig. R1Ac**). Using the CCH for estimating STG yielded inaccurate estimations (**Fig. R1Ac**).

To test whether deconvolution indeed improves STG/PSP estimation in the case of two bursting neurons, we first applied the algorithm to the CCH and ACHs of **Fig. R1Ac** and re-estimated the STG and PSP from the resulting dcCCH (**Fig. R1Ad**). Compared to the estimates derived from the CCH, all STG estimates based on the dcCCH were improved (**Fig. R1Ad**).

We then carried out simulations in which the presynaptic burst fraction was varied in an orderly manner (from 0 to 0.4 with 0.05 increments; $rSTG=0.04$; burst fraction of the postsynaptic neuron was fixed at 0.1; 30 repetitions each; other parameters were kept the same as in **Fig. R1A**). For every simulation, we estimated STGs and PSPs from the CCH and dcCCH using all five methods (**Fig. R1B**). We found that for the tails, jitter, and median predictors, eSTGs derived from the dcCCH were more accurate than the eSTGs derived from the CCH for all burst fractions ($p<0.001$, Mann-Whitney U -test). For the CoNNECT method, only the ePSPs derived from the CCH depended on the burst fraction (CoNNECT without deconvolution: $p=0.01$; with deconvolution: $p=0.053$; permutation test). For comparing STG and PSP-based estimates, we used the normalized slope of the best linear fit, after dividing all values by the mean estimates at zero burst fraction. Deconvolution reduced the effect of bursting for all methods (CoNNECT: $p=0.025$, $p<0.001$ for all other methods, bootstrap test; **Fig. R1C**). Thus, even when both neurons exhibit burst spiking activity, deconvolution improves STG and PSP estimation.

Following this comment, we added the results in **Fig. R1** to **Fig. 3** as new panels I-K (**Results**, pg. 13; and **Results**, pg. 14, lines 306-309), and extended the description in the text (**Results**; pp. 11-12, lines 269-282).

Figure R1. Deconvolution enables accurate estimation of spike transmission gain when both neurons exhibit burst spiking activity

A. Spike trains of two neurons coupled by an excitatory monosynaptic connection were simulated. The presynaptic and the postsynaptic spike trains exhibit bursting activity. “Burst fraction” is BR₁=0.4 for the presynaptic neuron (**Aa**) and BR₂=0.1 for the postsynaptic neuron (**Ab**). (**c**) In addition to a peak within the causal ROI ($0 < \tau \leq 5$ ms), the CCH exhibits prominent side lobes which are due to the burst spiking of the two neurons. (**d**) A deconvolved CCH (dcCCH; green) is derived from the ACHs and CCH (panels **Aa-c**). The dcCCH does not contain burst-induced side lobes, allowing to recover the unidirectional spike transmission curve STC₂₁ used in the simulation.

B. Spike trains coupled by an excitatory monosynaptic connection were simulated while modifying BR₁. BR₂ was kept constant at 0.3. (**a**) eSTG-to-rSTG ratio as a function of BR₁ for the three STG estimation methods using CCHs (solid lines) and using dcCCHs (dashed lines). In the presence of bursting, deconvolution improves STG estimation. (**b**) Estimated PSP as a function of burstiness for the CoNNECT and GLMCC methods, using CCHs and dcCCHs.

C. Normalized slope of the best linear fit of the curves in panel **B**. Error bars, SEM. Deconvolution reduces the effect of bursting for all methods (*/***: $p < 0.05/p < 0.001$, bootstrap test).

Comment #2

The above concern arises because burst spiking plays an important role in high level functions. So, bursts as artifacts... is that always justified? In this connection, the following reference appears to indicate that burst spiking in presynaptic neurons causes spiking in postsynaptic neurons.

Zeldenrust, Fleur, Wytse J. Wadman, and Bernhard Englitz. "Neural coding with bursts—current state and future perspectives." *Frontiers in computational neuroscience* 12 (2018): 48. <https://www.frontiersin.org/articles/10.3389/fncom.2018.00048/full>

Or, is this a characteristic of neuronal responses from the CA1 area in the hippocampus only? Please clarify

Addressing comment #2

Indeed, the importance of bursts has been studied extensively (Connors and Gutnick, 1990; Lisman, 1997; Izhikevich et al., 2003; Zeldenrust et al., 2018) and bursts are by no means artifacts. In hippocampal region CA1, a single burst can produce long-term potentiation (Thomas et al., 1998), and spike transmission between PYR-INT pairs in CA1 is higher for spikes which occur during bursts, compared to single spikes (Csicsvari et al., 1998). Hence, burst spiking activity is clearly important for spike transmission. We agree that the term “burst spiking artifact” was misleading.

When computing CCHs, researchers are interested in measuring the transmission of a single spike between two neurons. Therefore, we focused on estimating the transmission of a single spike, and showed that when spikes occur within bursts, the CCH exhibits side-lobes which cause an inaccurate estimation of the STG even when using state-of-the-art methods. In other words, the effect of a single spike can be differentiated from the effect of bursts. We would like to clarify that we used the term “burst spiking artifact” only in this context, meaning that bursts activity interferes with the measurement of the STG. Deconvolution removes the effect of the bursts on the STG, enabling the estimation of spike transmission per a single spike.

Following this comment, we removed the misleading terms from the text. Specifically, we made the following changes:

We changed **Abstract, pg. 1, lines 14-15** to read “**Deconvolution removed the effect of burst spiking, improving the estimation of neuronal connectivity yielded by state-of-the-art methods.**”

We changed **Introduction, pg. 3, lines 63** to read “Using deconvolution in concert with timescale separation methods removed **the effect of burst spiking**, improving the accuracy of both detection and quantification of neuronal connectivity in synthetic data.”

We changed **Results, pg. 10, line 198** to read “Deconvolution eliminates burst spiking **effects** and enables accurate STG and PSP estimation.”

We changed **Discussion, pg. 22, line 507** to read “Deconvolution removed the **effect of burst spiking** from the CCH and recovered the constructed connectivity.”

References:

Connors, B. W. & Gutnick, M. J. Intrinsic firing patterns of diverse neocortical neurons. *Trends in Neurosciences* 13, 99–104 (1990).

Csicsvari, J., Hirase, H., Czurko, A. & Buzsáki, G. Reliability and state dependence of pyramidal cell-interneuron synapses in the hippocampus: an ensemble approach in the behaving rat. *Neuron* 21, 179-189 (1998).

Izhikevich, E. M., Desai, N. S., Walcott, E. C. & Hoppensteadt, F. C. Bursts as a unit of neural information: selective communication via resonance. *Trends in Neurosciences* 26, 161–167 (2003).

Lisman, J. E. Bursts as a unit of neural information: making unreliable synapses reliable. *Trends in Neurosciences* 20, 38–43 (1997).

Soltész, I., Smetters, D. K. & Mody, I. Tonic inhibition originates from synapses close to the soma. *Neuron* 14, 1273–1283 (1995).

Thomas, M. J., Watabe, A. M., Moody, T. D., Makhinson, M. & O'Dell, T. J. Postsynaptic Complex Spike Bursting Enables the Induction of LTP by Theta Frequency Synaptic Stimulation. *Journal of Neuroscience* 18, 7118–7126 (1998).

Zeldenrust, F., Wadman, W. J. & Englitz, B. Neural Coding With Bursts—Current State and Future Perspectives. *Frontiers in Computational Neuroscience* 12, 48 (2018).

Comment #3

While CCH is defined, ACH is not.

Addressing comment #3

Following this comment, we added the definition for the ACH in the text (Results, pg. 7, lines 157-158).

Comment 4

Figures should be presented in better quality. Font size must be increased (difficult to read at present).

Addressing comment #4

Following this comment, font size of all figures is now at least 6.

Comment 5

Why a bin size of 1 ms? Please clarify, possibly in relation to the following reference. Carro-Martín, Carmen Rocío, José M. Delgado-García, Agnes Gruart, and R. Sánchez-Campusano. "Spike

sorting based on shape, phase, and distribution features, and K-TOPS clustering with validity and error indices." *Scientific reports* 8, no. 1 (2018): 1-28.

Addressing comment #5

A bin size of 1 ms was chosen for two reasons. First, to maintain consistency with the evaluation of the GLMCC (Kobayashi et al., 2019) and the CoNNECT (Endo et al., 2021) methods. The CoNNECT method can operate only on CCH with bin size of 1 ms. Second, a 1 ms bin size has been used in multiple previous studies: extracellular (Csicsvari et al., 1998), intracellular (Jouhanneau et al., 2018), and across species (Dickey et al., 2021). This is because 1 ms is, under most circumstances, the largest bin size which still ensures that two consecutive spikes generated by the same neuron will not be allocated to the same bin, keeping the counting process Poisson. To meet this condition, the minimal inter-spike interval should be larger than the bin size. For both the point process and the conductance-based simulations we ensured an absolute refractory period (ARP) of at least 1 ms. For the real dataset, we used only PYR and INT recorded from CA1; for these neurons, the extracellular spike width is always above 0.6 ms and the ARP is several ms (Stark et al., 2013). Notably, when spike rate is very high, ARP is very low, and spike duration is shorter than 0.5 ms (Caro-Martín et al., 2018), two consecutive spikes emitted by the same neuron might be allocated to the same bin. Then, smaller bin sizes should be employed.

Nevertheless, deconvolution can be used to produce dcCCHs which are clean from second order spike train statistics regardless of the CCH bin size. For demonstration, we simulated two spike trains coupled by an excitatory monosynaptic connection ($rSTG=0.04$; 240 min; **Fig. R2A**). The spike train of the presynaptic neuron was initially simulated as a Poisson process modified by refractoriness ($ARP_1=2$ ms) and exhibited bursting activity ($BR=0.4$). The spike train of the postsynaptic neuron realized a second-order gamma process modified by refractoriness ($ARP_2=2$ ms). Three CCHs were computed using bin sizes of 0.2, 0.5, and 1 ms. Consistent with the construction, all three CCHs exhibited peaks within the causal ROI and side lobes, due to the burst spiking activity (**Fig. R2A**). Following deconvolution, in all three dcCCH the side lobes were removed, and the eSTGs yielded by the dcCCH were more accurate than the eSTGs produced by median filter without deconvolution (**Fig. R2B**). Next, we computed CCHs using 0.2, 0.5, and 1 ms bin sizes for a real pair of spike trains recorded from the CA1 (**Fig. 6C**, third pair from the left). The CCHs exhibited peaks within the causal ROI and side lobes (**Fig. R2C**). Applying deconvolution to the real data CCHs produced dcCCHs without side lobes, and eSTGs higher than those yielded by median filtering without deconvolution.

Following this comment, we now explain the rationale for using a bin size of 1 ms (Methods, pg. 25, lines 615-618; References, pg. 44, lines 1192-1194).

References:

Caro-Martín, C. R., Delgado-García, J. M., Gruart, A. & Sánchez-Campusano, R. Spike sorting based on shape, phase, and distribution features, and K-TOPS clustering with validity and error indices. *Scientific Reports* 8, 17796 (2018).

Csicsvari, J., Hirase, H., Czurko, A. & Buzsáki, G. Reliability and state dependence of pyramidal cell-interneuron synapses in the hippocampus: an ensemble approach in the behaving rat. *Neuron* 21, 179-189 (1998).

Dickey, C. W. et al. Travelling spindles create necessary conditions for spike-timing-dependent plasticity in humans. *Nature Communications* 12, 1027 (2021).

Endo, D. et al. A convolutional neural network for estimating synaptic connectivity from spike trains. *Sci Rep* 11, 12087 (2021).

Jouhanneau, J. S., Kremkow, J. & Poulet, J. F. A. Single synaptic inputs drive high-precision action potentials in parvalbumin expressing GABA-ergic cortical neurons in vivo. *Nature Communications* 9, 1540 (2018).

Kobayashi, R. et al. Reconstructing neuronal circuitry from parallel spike trains. *Nature Communications* 10, 4468 (2019).

Stark, E., Eichler, R., Roux, L., Fujisawa, S., Rotstein, H.G., & Buzsáki, G. Inhibition-Induced Theta Resonance in Cortical Circuits. *Neuron* 80, 1263–1276 (2013).

Figure R2. Deconvolution removes second order statistics of simulated and real CCHs with different bin sizes

A. Spike trains of two neurons coupled by an excitatory monosynaptic connection were simulated. The presynaptic spike train exhibited bursting activity ($BR=0.4$). (a-c) Raw CCHs computed with bin sizes of 0.2, 0.5, and 1 ms. All CCHs exhibit side lobes, due to the burst spiking of the presynaptic neuron.

B. Deconvolved CCHs (dcCCH) derived from the CCHs shown in panel A. The dcCCHs are free from the effect of spike bursts, and yield more accurate estimates of the STG.

C. CCHs computed with 0.2, 0.5, and 1 ms bin sizes for real spike train data recorded from CA1 (same example as in Fig. 6C, third pair from left). All three CCHs exhibit side lobes.

D. dcCCHs for the data presented in C. The dcCCHs all yield higher eSTGs than those obtained from the CCHs (panel C).

Comment #6

How did you choose the temporal region (time interval) of interest as [0 5] ms while finding STC?

Addressing comment #6

The temporal region of interest (ROI) was chosen to ensure including at least part of the STC. The actual value is based on two parameters: the delay and the jitter of spike transmission. A short delay of spike transmission and a narrow STC allow focusing on a short ROI near zero lag. Previous work in neocortex (Fujisawa et al., 2008) and hippocampus showed a peak delay between 0-3 ms (English et al., 2017; Gridchyn et al., 2020; Adeyelu et al., 2022; Csicsvari et al., 1998). Consistent with this literature, we focused on spike transmission between directly connected neurons, i.e., on monosynaptic connections.

Following this comment, we now explain the rationale for using an ROI of [0 5] (Methods, pg. 26, lines 650-654).

References:

Adeyelu, T., Shrestha, A., Adeniyi, P. A., Lee, C. C. & Ogundele, O. M. CA1 Spike Timing is Impaired in the 129S Inbred Strain During Cognitive Tasks. *Neuroscience* 484, 119–138 (2022).

Csicsvari, J., Hirase, H., Czurko, A. & Buzsáki, G. Reliability and state dependence of pyramidal cell-interneuron synapses in the hippocampus: an ensemble approach in the behaving rat. *Neuron* 21, 179-189 (1998).

English, D. F. et al. Pyramidal Cell-Interneuron Circuit Architecture and Dynamics in Hippocampal Networks. *Neuron* 96, 505-520 (2017).

Fujisawa, S., Amarasingham, A., Harrison, M. T. & Buzsáki, G. Behavior-dependent short-term assembly dynamics in the medial prefrontal cortex. *Nature Neuroscience* 11, 823–833 (2008).

Gridchyn, I., Schoenenberger, P., O’Neill, J. & Csicsvari, J. Optogenetic inhibition-mediated activity-dependent modification of CA1 pyramidal-interneuron connections during behavior. *eLife* 9, e61106 (2020).

Comment #7

In Figure 5B(b), we see inhibitory conductance $G_{ei} = 0.25$. It is 0.025 in the description. Please rationalize.

Addressing comment #7

We apologize for the typo and contradiction. The correct value is 0.25 mS/cm^2 , as was noted in **Fig. 5Bb**.

Following this comment, the text was fixed to $G_{ei} = 0.25 \text{ mS/cm}^2$ (Results, pg. 17, line 401).

Reviewer #3**Comment #1**

In Eq. (7.2), the authors claim that the impulse response from neuron 2 to 1, h_2 is non-zero for negative time lags, and that is the ‘causal direction’ from 2 to 1. Since this is not a very standard definition of causality, it would be helpful if this could be explained in more detail.

Addressing comment #1

We agree. Following this comment, we modified the definition of h_2 to support the standard definition of causality (Methods, pg. 27, lines 683-685).

Comment #2

In Eq. (12), the denominator should be corrected as $|1-H_1(f)H_2(f)|^2$, since it is a product of a complex number and its conjugate. This error has then propagated to Eq. (14) as well.

Addressing comment #2

Agreed. Fixed (Methods, pg. 28, lines 706 and 714).

Comment #3

The authors claim that the last element and the denominator of Eq. (14) vanished if the feedback loop: $H_1(f)H_2(f)$ is small. For both of those terms to vanish, both $H_1(f)H_2(f)$ and $H_1(f)\bar{H_2(f)}$ need to go to zero. The only way that both of these quantities seem to go to zero is if at least one of $H_1(f)$ or $H_2(f)$ independently goes to zero. It would be better if the authors can elaborate the settings that these assumptions hold and does not hold in practice, for the benefit of the future users of these methods.

Addressing comment #3

We agree that $H_1(f)H_2(f)$ will be exactly zero if at least one of $H_1(f)$ or $H_2(f)$ is zero. However, if both are small, the product will be even smaller than either one, and could be ignored.

In general, the product of $H_1(f)$ and $H_2(f)$ can be ignored when the loop gain is lower than a minimal unidirectional STG, which can be identified with finite data. The detection of a connection depends on multiple variables including recording duration, firing rates, and the method employed. Therefore, an exact number for a minimal detectable unidirectional STG cannot be obtained. However, in the specific case of unidirectional connection with an impulse response spanning a single bin and Poisson firing pattern of both neurons, the minimal detectable unidirectional STG can be computed exactly using the tails predictor. Under these conditions, the CCH baseline is

$$\lambda = F_1 F_2 T B$$

where F_1 and F_2 are the firing rates of the pre- and postsynaptic neurons in spk/s, T is the effective recording duration, and B is the CCH bin width (both measured in s; Stark and Abeles, 2009). For a given λ and alpha (detection threshold) level, the inverse Poisson distribution can be used to obtain the minimal number of transmitted spikes required for detection. The minimal STG is then calculated as the minimal number of transmitted spikes, divided by $F_1 T$.

To give a general range of what is a minimal unidirectional STG which can be identified based on finite data, we calculated the minimal STG for ranges of $F_1 F_2$ at $T = 50,000$ s (~ 14 hours), $B = 0.001$ s, and an alpha level of 0.001. For F_1 we used a range of 0.5-3 spk/s, and for F_2 we used a range of 1-30 spk/s. Under these conditions, the range of minimal detectable unidirectional STG is between 0.00026 and 0.00345 (**Fig. R3**). Thus, in the general case of bidirectional connectivity, when the loop gain is below 0.00026-0.00345, the product elements are effectively zero and **Eq. 15** is likely to hold. The maximal “safe” loop gain corresponds to symmetric connections with unidirectional STGs 0.016-0.058.

Figure R3. Minimal STG that can be detected in finite data

Minimal STG were calculated for different firing rates of the presynaptic neuron F_1 (different lines) as a function of the firing rate of the postsynaptic neuron, F_2 . Recording duration and bin size were kept constant at $T = 50,000$ s and $B = 0.001$ s. The firing rate of the presynaptic neuron is shown on the left side for each line.

To further demonstrate the effect of weak and strong reciprocal connectivity on deconvolution outcome, we simulated two neurons connected by weak (rSTG=0.01, within the “safe” range of loop gain as calculated above) and strong (rSTG=0.09) excitatory bidirectional monosynaptic connections. Both spike trains were initially simulated as Poisson processes modified by refractoriness ($\lambda_1=3$ spk/s; $\lambda_2=3$ spk/s; ARP=2 ms). The presynaptic neuron exhibited bursting

(BR=0.4). For the weak connection, the side lobes apparent in the CCH were removed following deconvolution (**Fig. R4Ac-d**). For the strong connection, the side lobes in the CCH were also removed, but the dcCCH exhibited distortion, apparent as negative side lobes (two dips on the left sides of the excitatory peaks; **Fig. R4Bc-d**). This distortion is consistent with the violation of the negligible loop gain assumption inherent in deriving **Eq. 15**.

Following this comment, we now explain how to determine whether a given pair of neurons is so “strongly coupled” that **Eq. 15** should not be used (**Methods, pg. 29, lines 736-750**).

Figure R4. Weak and strong excitatory reciprocal connections

A. Spike trains of two neurons with reciprocal excitatory connections ($rSTG_{12}=rSTG_{21}=0.01$) were simulated. The spike train of neuron 1 exhibited burst spiking activity (BR=0.4), whereas the spike train of neuron 2 did not exhibit bursts. (a) ACH of neuron 2. (b) ACH of neuron 1. (c) CCH between the two neurons. (d) dcCCH for the CCH in c.

B. Same as **A**, with reciprocal excitatory connections of $rSTG=0.09$.

References:

Stark, E. & Abeles, M. Unbiased estimation of precise temporal correlations between spike trains. *Journal of Neuroscience Methods* 179, 90-100 (2009).

Comment #4

In Eq. (15.1) and (16.1), the second term on the RHS should be corrected as $F(\text{ACH}_1^0)\bar{H}_1(f)$. This error then propagates to all subsequent steps. For example, in Eq. (15.2) and (16.2), the second term on the RHS should be corrected as $\text{ACH}_1^0 * h_1$. Further, the numerator of the RHS of Eq. (18.1) should also be corrected as $\bar{H}_1(f)$.

Addressing comment #4

Agreed. As noted above, following Comment #1 of Reviewer #3 we modified the definition of h_2 to support the standard definition of causality. To maintain consistency, we also updated the Fourier transform of the CCH in **Eq. 11.2** and the CCH definitions in **Eq. 13** (**Methods, pg. 28, lines 702-703 and 710**).

Following these modifications, conjugation is applied to $S_1(f)$, and **Eq. 15.1** has been modified accordingly. Furthermore, **Eq. 15.2** has been duly corrected by time-reversing h_2 . Following this comment, **Eq. 15**, and **Eq. 18**, **Eq. 19**, **Eq. 21**, and the associated text were modified (**Methods, pg. 28, lines 718, 720, 726, 731; Methods, pg. 29, lines 765 and 768; Methods, pg. 30, line 773**).

Comment #5

Then, since \tilde{h}_1 is a time reversed version of h_1 (because of the conjugation mentioned above), \tilde{h}_1 and \tilde{h}_2 are going to lie in the same side of the zero lag (since the authors originally claimed that h_1 and h_2 reside on the opposite sides). This makes the statement in Page 27: line 721 to be irrelevant. It is also noteworthy that had the authors originally claimed that both h_1 and h_2 are non-zero only for positive lags (which is the standard notion of causality), \tilde{h}_1 and \tilde{h}_2 would have actually laid on the opposite sides of the zero lag.

Addressing comment #5

We thank the Reviewer for this comment, which we believe helped to clarify the exposition.

Following this and other comments (#1, #2, and #4 of Reviewer #3), we modified **Eq. 19** and **Eq. 21** to represent the contribution of the two impulse responses on different sides of the zero lag (**Methods, pg. 29, line 768; and Methods, pg. 30, lines 773**).

Comment #6

A fundamental assumption in this derivation is that if the number of transmitted spikes is small, then $\text{ACH}_1^0 \sim \text{ACH}_1$ and $\text{ACH}_2^0 \sim \text{ACH}_2$. For potential reproducibility and future usage, it is important to know what settings are required to meet these assumption. Thus, it would be useful if the authors can give an idea about the practical settings where these assumptions hold true, and if possible demonstrate through simulations cases where their assumptions are no longer valid.

Addressing comment #6

Indeed, when ACH_2^0 is not approximately equal to ACH_2 , **Eq. 15** cannot be evaluated and the outcome of the deconvolution algorithm (**Eq. 21**) will be distorted.

The spike train of neuron 2, $s_2(t)$, is a superposition of the background spike train and the transmitted spike train:

$$s_2(t) = s_2^0(t) + \int_{-\infty}^{\infty} s_1(t - \tau)h_1(\tau)d\tau$$

Thus, when the transmitted spike train is Poisson and has a flat (delta function) auto-correlation, there will be no differences in the shape of the postsynaptic ACH, and the scaled ACH_2 will be identical to the scaled ACH_2^0 . However, when the transmitted train is non-Poisson (which can happen if the presynaptic train, $s_1(t)$, is itself non-Poisson), the outcome of the deconvolution algorithm will be distorted.

Thus, whenever the presynaptic spike train is Poisson (i.e., ACH_1 is flat), the shape of ACH_2 will remain the same as ACH_2^0 . Furthermore, whenever the number of transmitted spikes is small relative to the number of background spikes (e.g., when the impulse response in the given direction is small), the deviation of ACH_2 with respect to ACH_2^0 will be small, even when ACH_1 is not flat.

To obtain a quantitative demonstration of how transmitted spikes affect the ACH_2 and thereby the outcome of the deconvolution algorithm, we simulated two neurons connected by an excitatory mono-synapse. In each simulation, the rSTG was varied between zero and 0.84. Both spike trains were initially simulated as Poisson processes modified by refractoriness ($\lambda_1=10$ spk/s; $\lambda_2=3$ spk/s; ARP=2 ms), and the presynaptic neuron exhibited bursting activity (BR=0.4). When rSTG was zero (no connection at all), the shape of ACH_2 remained Poisson (**Fig. R5Aa**), and the CCH (**Fig. 5Ba**) and the dcCCH were similar (**Fig. R5Ca**). When the rSTG was set to a very high value (0.228) but below unity, the shape of ACH_2 deviated from Poisson (**Fig. R5Ab**). The CCH exhibited bursting side lobes, and the eSTG/rSTG was 0.69 (**Fig. R5Bb**). However, the dcCCH did not exhibit bursting side lobes, and the eSTG/rSTG was close to one (eSTG=1.02; **Fig. R5Cb**).

For an rSTG near unity (0.84), the ACH_2 exhibited a burst shape inherited by the transmitted spikes of the presynaptic neuron (**Fig. R5Ac**). The CCH exhibited burst side lobes, and the eSTG/rSTG was 0.78 (**Fig. R5Bc**). The dcCCH exhibited negative side lobes (dips on both sides of the monosynaptic peak; **Fig. R5Cc** inset), consistent with the strong deviation of ACH_2 from ACH_2^0 . Nevertheless, the eSTG/rSTG given by the dcCCH was improved at 1.04 (**Fig. R5Cc**). However, the left negative side lobe was falsely detected as an inhibitory connection. Thus, when the rSTG is high and the transmitted spikes exhibit burst activity, the shape of ACH_2 is not flat even when ACH_2^0 is flat, potentially distorting the outcome of the deconvolution process.

To quantify the effect of bursting transmitted spikes on ACH_2 , we calculated the Euclidian distance between the normalized ACH_2^0 (the ACH of the postsynaptic neuron before adding the transmitted spikes) and the normalized ACH_2 (the ACH of the postsynaptic neuron after adding the transmitted spikes). When the rSTG is zero, ACH_2 is identical to ACH_2^0 and the Euclidian distance between the normalized ACH_2^0 and the ACH_2 is zero (**Fig. R5D**). For non-zero rSTG, the distance between the ACHs increases with rSTG ($p<0.001$, permutation test).

To determine the conditions under which a deviation of ACH_2 from ACH_2^0 affects spike transmission estimation and detection, we estimated the STG in the null direction ($eSTG_{12}$; $rSTG_{12}=0$). When the $rSTG$ was below 0.25, the $eSTG_{12}$ yielded by the dcCCH was close to zero, and no false detection occurred (**Fig. R5E, green**). For $rSTGs$ above 0.25, the absolute values of $eSTGs$ in the null direction were higher, and were falsely detected as inhibitory connections. Estimations of $STGs$ in the null direction given by the CCH were falsely detected as excitatory connections for $rSTG$ from 0.64 (**Fig. R5E, grey**).

Finally, we computed the $eSTG/rSTG$ ratio in the connected direction, for $eSTGs$ produced by the CCH and the dcCCH. Overall, the $eSTGs$ produced by the dcCCH were more accurate than $STGs$ based on the CCH ($p<0.001$, U -test; **Fig. R5F**).

Thus, when the shape of ACH_2 highly deviates from the ACH_2^0 , the dcCCH exhibits negative side lobes which distort the deconvolution outcome. However, this distortion has an actual effect only when (1) the number of transmitted spikes is high relative to the baseline spikes and (2) the transmitted spikes exhibit burst activity ($BR=0.4$).

Following this comment and the ensuing analyses, we now explain that as long as the number of transmitted spikes is small relative to the number of background spikes, even when the transmitted spikes are bursting, the approximation of **Eq. 17** holds for all practical purposes. Furthermore, we describe the settings in which these assumptions hold true. Therefore, made the approximation in **Eq. 17** explicit (**Methods, pp. 29, lines 757-758**) and added the following text (**Methods, pg. 30, lines 791-802**):

“The deconvolution algorithm as derived above (**Eq. 17-21**) is based on **Eq. 15**, but makes two key simplifying assumptions. First, that the shape of the ACH is not considerably distorted by the transmitted spikes (**Eq. 17**). Second, that the impulse response is not considerably distorted by the divisive scaling factor inherent in the derivation (**Eq. 18**).

With respect to **Eq. 17**, using point process simulations, we found that even when the presynaptic neuron spikes at a relatively high rate (10 spk/s) with strong bursting activity ($BR=0.4$), and the postsynaptic neuron spikes at a lower rate (3 spk/s), the estimates of connectivity are not distorted as long as the $rSTG$ is below 0.25. Notably, strong connections and high firing rates are not characteristic of local cortical networks. However, different settings may appear in different systems, e.g., thalamocortical feedforward connections (Bruno and Sakmann, 2006). When connectivity is very strong and presynaptic firing rates are high, prior knowledge about connectivity can be used to employ unidirectional deconvolution (**Eq. 21.3-4**) instead of the “standard” deconvolution (**Eq. 21.1-2**).”

Figure R5. Deconvolution recovers constructed connectivity even when high STGs change the ACH of the postsynaptic neuron

Spike trains of two neurons with a monosynaptic excitatory connection were simulated. In each simulation, the rSTG was varied. The spike trains of the pre- and postsynaptic neurons were initially simulated as Poisson processes, modified by refractoriness. The BR of the presynaptic neuron was set to 0.4; BR of the postsynaptic neuron was zero.

A. Example ACHs of the postsynaptic neuron at rSTG=0/0.228/0.839 (a/b/c). When rSTG is close to unity (e.g., c), the spike train of the postsynaptic neuron exhibits bursting, which is entirely inherited from the presynaptic train.

B. CCHs for the same simulations as in A.

C. dcCCH for the CCHs in B. When rSTG is very high (nearly one-to-one transmission), deconvolution generates two dips, on the two sides of the monosynaptic peak (c, inset).

D. For each simulation, Euclidian distances were calculated between the normalized postsynaptic ACHs, ACH_2^0 and the ACH_2 , and plotted against the rSTG.

E. eSTG in the null direction (eSTG₁₂) plotted against the rSTG, for eSTGs derived from the raw CCH (brown) and for eSTGs derived from the dcCCH (green). At high rSTGs (rSTG>0.25; vertical green dashed line), eSTGs in the null direction derived from the dcCCH are falsely detected as inhibitory. For rSTG ≥ 0.64 (vertical brown dashed line), the eSTG in the null direction derived from the CCH are falsely detected as excitatory.

F. eSTG/rSTG plotted against the rSTG, for eSTGs derived from the raw CCH (brown) and for eSTGs derived from the dcCCH (green). At high rSTGs (rSTG>0.25), deconvolution overestimates the rSTGs. However, estimates yielded by deconvolution are more accurate than estimates based on the raw CCH (without deconvolution) for all tested rSTGs (three orders of magnitude, 0.001-1).

References:

Bruno, R. M. & Sakmann, B. Cortex is driven by weak but synchronously active thalamocortical synapses. *Science* 312, 1622–1627 (2006).

Comment #7

While it is true that the multiplicative scaling of the filters $H_1(f)$ and $H_2(f)$ by the ACHs is eliminated by the deconvolution proposed in Eq. (21.2), it is also important to note that this is achieved at the cost of introducing a new divisive scaling factor. After this deconvolution, the recovered variables are \tilde{h}_1 and \tilde{h}_2 , which are again scaled versions of the desired quantities h_1 and h_2 . It would be great if the authors could justify why this newly added scaling factor is not consequential. This is very important because the authors claim that the ‘deconvolution results in a CCH which is free from ACH artifacts’ (e.g. Page 10: line 231,234), which does not seem to be completely true according to their derivation.

Addressing comment #7

The divisive factor in **Eq. 18** deviates from unity whenever the corresponding scaled ACH deviates from a delta function, which happens whenever the corresponding spike train is non-Poisson. When the corresponding spike train exhibits Poisson or higher gamma order spiking activity, the scaled ACH remains similar to a delta function and the new scaling factor is not consequential for all practical purposes.

Specifically, if connectivity is unidirectional and the postsynaptic train is Poisson (i.e., the scaled ACH_2 is an approximation of a delta function), then the estimate of the connectivity will remain undistorted even if the presynaptic train is non-Poisson, since $\tilde{H}_1(f) = \frac{H_1(f)}{\mathcal{F}(ACH_2)}$. Furthermore, if connectivity is unidirectional (i.e., $H_2(f) = 0$) and the presynaptic train is non-Poisson, the estimate of the null direction impulse response will not be distorted, since the numerator of the r.h.s. of $\tilde{H}_2(f) = \frac{H_2(f)}{\mathcal{F}(ACH_1)}$ is zero. When connectivity is bidirectional and one of the trains is non-Poisson (**Fig. R4**), or when connectivity is unidirectional and the postsynaptic train is non-Poisson, the scaling factor may become consequential. To avoid distortion by the divisive scaling factor, prior knowledge on the connectivity can be used to apply one-sided deconvolution.

To demonstrate the effect of the divisive scaling factor, we simulated two neurons connected by an excitatory monosynaptic connection (rSTG=0.04). In each simulation, burst fraction of the

postsynaptic neuron was varied between zero and 0.4. Both spike trains were initially simulated as Poisson processes modified by refractoriness ($\lambda_1=10$ spk/s; $\lambda_2=3$ spk/s; ARP=2 ms); the presynaptic spike trains did not exhibit bursting (BR=0; **Fig. R6Aa-b**). The CCH, dcCCH, and one-sided dcCCH (deconvolution of the CCH with the ACH of the presynaptic neuron; see below) were similar, and the eSTG/rSTG was 0.96 for the CCH (**Fig. R6Ac**), 1.02 for the dcCCH (**Fig. R6Ad**) and 1 for the one-sided dcCCH (**Fig. R6Ae**). The small difference between eSTGs of the CCH and dcCCHs is due to the fact that the scaled ACH_2 is not flat (i.e., is not exactly a delta function).

We then repeated the simulation with same parameters, with the addition of postsynaptic bursting activity (BR=0.4; **Fig. R6Ba**). The CCH and the one-sided dcCCH were flat on both sides of the monosynaptic peak, and the eSTG/rSTG for the CCH and the one-sided dcCCH were 0.958 and 1.004, respectively (**Fig. R6Bc; Fig. R6Be**). The dcCCH exhibited negative side lobes on both sides of the monosynaptic peak, and the eSTG/rSTG was 1.07 (**R6Bd**).

To quantify the effect of burst spiking activity on deconvolution outcome, we carried out simulations with same parameters as in **Fig. R6AB**. In each simulation, we modified the postsynaptic burst fraction between zero and 0.4. We then calculated the eSTGs in the null direction (eSTG₁₂; rSTG₁₂=0) for the CCH, the dcCCH, and the one-sided dcCCH. Consistent with the examples (**Fig. R6AB**), eSTGs in the null direction given by the dcCCH were less accurate than eSTGs given by the CCH, and the one-sided dcCCH (p<0.001 *U*-test; **Fig. R6C**). Next, we computed the eSTG/rSTG for CCH, dcCCH, and one-sided dcCCH. We found that the eSTGs produced by the one-sided dcCCH were more accurate than the eSTGs given by the CCH and dcCCH (p<0.001 *U*-test; **Fig. R6D**).

Thus, when the corresponding ACH exhibits Poisson or high order gamma activity, the scaling factor is negligible for all practical reasons. However, when the scaled ACH is not identical to a delta function, a scaling factor that deviates from unity is indeed added. When the corresponding ACH exhibits burst activity, the scaling factor can introduce negative side lobes.

To formalize the extension of the deconvolution method in such cases, denoted above as “one-sided deconvolution”, we proceed as follows. Given prior knowledge about unidirectional information flow in a system, **Eq. 15**

$$\mathcal{F}(CCH_{12}) = \mathcal{F}(CCH_{12}^0) + \mathcal{F}(ACH_1^0)H_1(f) + \mathcal{F}(ACH_2^0)\overline{H_2(f)} \quad (15.1)$$

$$CCH_{12} = s_1^0 * s_2^0 + ACH_1^0 * h_1 + ACH_2^0 * h_2(-\tau) \quad (15.2)$$

Can be replaced by **Eq. 16**:

$$\mathcal{F}(CCH_{12}) = \mathcal{F}(CCH_{12}^0) + \mathcal{F}(ACH_1^0)H_1(f) \quad (16.1)$$

$$CCH_{12} = s_1^0 * s_2^0 + ACH_1^0 * h_1 \quad (16.2)$$

Then, the deconvolution algorithm can be simplified, and **Eq. 21.1-2**:

$$\tilde{h}_1 + \tilde{h}_2(-\tau) + OI = \mathcal{F}^{-1} \left[\frac{\mathcal{F}(CCH_{12})}{\mathcal{F}(ACH_1)\mathcal{F}(ACH_2)} \right] \quad (22.1)$$

$$dcCCH_{12} = \mathcal{F}^{-1} \left[\frac{\mathcal{F}(CCH_{12})}{\mathcal{F}(ACH_1)\mathcal{F}(ACH_2)} \right] \quad (22.2)$$

Can be replaced by the new **Eq. 21.3-4**:

$$h_1 + OI = \mathcal{F}^{-1} \left[\frac{\mathcal{F}(CCH_{12})}{\mathcal{F}(ACH_1)} \right] \quad (22.3)$$

$$dcCCH_{12} = \mathcal{F}^{-1} \left[\frac{\mathcal{F}(CCH_{12})}{\mathcal{F}(ACH_1)} \right] \quad (22.4)$$

As noted above, when one-sided deconvolution is applied to the data simulated in **Fig. R6CD**, the distortions in the null-direction disappear (**Fig. R6**). We thank the Reviewer for making this comment, that we believe helped expand the applicability of deconvolution.

Following this comment, we added **Eq. 21.3** and **21.4** (**Methods**, pg. 30, lines 777-779). Second, we modified the sentence on pg. 10, lines 233 to read “...resulting in a CCH which does not exhibit ACH-induced side lobes...” Furthermore, we added the following text (**Methods**, pg. 31, lines 803-811):

“While ACHs effects are eliminated by the deconvolution algorithm, a new divisive scaling factor (the denominator of **Eq. 18**) is introduced which can distort the recovered impulse responses. If connectivity is unidirectional and the postsynaptic train is Poisson, the scaled ACH_2 is a delta function, the divisive factor in **Eq. 18.1** is unity, and $\tilde{h}_1 = h_1$. Furthermore, if connectivity is unidirectional, then even if the presynaptic train is non-Poisson, the numerator of the r.h.s. of **Eq. 18.2** is zero and thus $h_2 = 0$. When connectivity is bidirectional and one of the trains is non-Poisson, or when connectivity is unidirectional and the postsynaptic train is non-Poisson, the divisive factor may become consequential. Then, prior knowledge about connectivity can be used to employ unidirectional deconvolution (**Eq. 21.3-4**).”

Figure R6. Effect of the divisive scaling factor on deconvolution outcome and a practical solution in relevant cases

A. Spike trains of two neurons with monosynaptic excitatory connections were simulated ($rSTG=0.04$). The spike trains of both neurons were initially simulated as Poisson processes modified by refractoriness. No bursts were simulated. **(a)** ACH of the postsynaptic neuron. Note slight deviation from Poisson due to transmitted spikes. **(b)** ACH of the presynaptic neuron. **(c)** CCH between the two neurons. **(d)** dcCCH for the CCH in **c**. **(e)** One-sided dcCCH.

B. Same as **A** with burst activity ($BR=0.4$) added to the postsynaptic spike train.

C. $eSTG_{12}$ in the null direction, for different burst fractions of the postsynaptic neuron. Horizontal dashed line at zero represents the STG in the null direction ($rSTG_{12}=0$).

D. $eSTG_{21}/rSTG_{21}$ in the connected direction, for different burst fractions of the postsynaptic neuron.

Comment #8

While the authors provide extensive simulations showing that deconvolution improves the connectivity estimates when the spike trains exhibit bursting activities, they also claim that the deconvolution can remove 'second order spike train statistics from the CCH' (Page 2: line 61, Page 10: line 228 and Page 20: line 484). It would be helpful if they can provide insights on settings other than bursting where their method yields better connectivity estimates.

Addressing comment #8

Indeed, deconvolution improves connectivity estimates for second order spike train statistics in general. While bursting is a relatively common deviation from Poisson spiking in real spike trains, other deviations are clearly possible. For instance, in some brain regions, strong periodicity is encountered, and deconvolution can be used to mitigate the distortion of connectivity estimates or even of slower features in the CCH.

To demonstrate how deconvolution can remove second order spike train statistics which are not bursts, we simulated two spike trains ($rSTG=0.05$, 166 min). The spike train of the presynaptic neuron did not exhibit bursting activity, but did exhibit strong ripple-band periodicity (150 Hz; **Fig. R7Aa**). The actual spiking of the presynaptic neuron was 20 spikes/s, of which 80% were rhythmic and 20% were background. Thus, the ACH of the presynaptic neuron exhibited rhythmic activity. The spike train of the postsynaptic neuron initially realized a second-order gamma process. The spike trains of both neurons were modified by refractoriness ($ARP=2$ ms). The CCH between the two spike trains exhibited also rhythmic activity, and the $eSTG/rSTG$ given by the CCH was 0.877 (**Fig. R7Ac**). After deconvolution, the rhythmic features in the $dcCCH$ were reduced relative to the CCH, and the $eSTG/rSTG$ given by the $dcCCH$ was improved to 1.006 (**Fig. R7Ad**).

Next, we simulated two neurons with the same parameters as in **Fig. R7A**, changing the rhythmic frequency from the ripple to the gamma (40 Hz) range. The ACH of the presynaptic neuron and the CCH again exhibited rhythmic activity, with lower frequency (**Fig. R7Bb-c**), and the $eSTG/rSTG$ given by the CCH was 0.997. After deconvolution, the rhythmic CCH features were reduced and the $eSTG/rSTG$ given by the $dcCCH$ was 1.034 (**Fig. R7Bd**). Thus, although deconvolution did not have a substantial effect on $eSTG$ at 40 Hz, deconvolution did remove the rhythmic effects from the CCH.

Thus, the utility of deconvolution is not limited to removing the effects of burst activity or for estimating connectivity. Deconvolution can remove different second order statistics, thereby differentiating between the effect of single spikes and the general spike pattern.

Following this comment, we added the following text (Discussion, pp. 21-22, lines 534-543; References, pg. 43, lines 1149-1155):

"We showed that deconvolution improves connectivity estimates for bursting activity, but deconvolution can remove other effects of second order spike train statistics on the CCH. While bursting is a relatively common deviation from Poisson spiking in real spike trains, other deviations are clearly possible. For instance, in some brain regions, strong periodicity is encountered, for instance at the ripple (Buzsaki et al., 1992) or gamma (Gray et al., 1989) range. When individual neurons exhibit phase locking to field oscillations, the ACH may exhibit

periodicity (Gray et al., 1989; O’Keefe and Recce, 1993; Csicsvari et al., 1999). Then, the CCH between connected neurons will be affected by ACH periodicity. Deconvolution can be used to differentiate between the effect of transmitted spikes and the firing patterns of the presynaptic neuron. Thus, deconvolution can minimize the distortion of connectivity estimates and improve the analysis of slower CCH features.”

Figure R7. Deconvolution reduces the effect of oscillatory spike trains from the CCH

A. Spike trains of two neurons with monosynaptic excitatory connections were simulated ($rSTG=0.05$). The spike train of the pre- and postsynaptic neuron were initially simulated as Poisson processes modified by refractoriness. No bursts were simulated. A rhythmic train of 150 Hz (spiking at 16 spikes/s) with spike time jitter of 0.25 ms was added to the background Poisson spike train (4 spikes/s) of the presynaptic neuron. **(a)** ACH of the postsynaptic neuron. **(b)** ACH of the rhythmic presynaptic neuron. **(c)** CCH between the two neurons. **(d)** dcCCH for the CCH in **(c)**.

B. The presynaptic neuron exhibited rhythmic spiking at 40 Hz with jitter of 3 ms, and overall rate of 20 spikes/s. Other conventions are the same as in **A**. The eSTGs estimate the rSTG accurately for both the CCH **(c)** and the dcCCH **(d)**. Side lobes induced by rhythmicity at 25 ms from the monosynaptic peak are evident in the CCH but not in the dcCCH.

References:

Buzsáki, G., Horváth, Z., Urioste, R., Hetke, J., & Wise, K. High-frequency network oscillation in the hippocampus. *Science* 256, 1025–1027 (1992).

Csicsvari, J., Hirase, H., Czurkó, A., Mamiya, A., & Buzsáki, G. Oscillatory coupling of hippocampal pyramidal cells and interneurons in the behaving Rat. *Journal of Neuroscience* 19, 274–287 (1999).

Gray, C. M., König, P., Engel, A. K., & Singer, W. Oscillatory responses in cat visual cortex exhibit inter-columnar synchronization which reflects global stimulus properties. *Nature* 338, 334–337 (1989).

O'Keefe, J., & Recce, M. L. (1993). Phase relationship between hippocampal place units and the EEG theta rhythm. *Hippocampus* 3, 317–330.

Comment #9

Page 7: it would be helpful if the operators * and ★ are explicitly defined here as well. Also, it would be better if the abbreviation ACH is introduced in the text before the equation in line 159.

Addressing comment #9

Following this comment, we now explicitly define the operators * and ★ and introduce the ACH abbreviation (Results, pg. 7, lines 157, 158, and 164).

Comment #10

Page 12: Figure 3-A-a, please label the y axes and the individual subplots.

Addressing comment #10

Following this comment, we labelled all y axes and individual subplots in **Fig. 3Aa-b**.

Comment #11

Page 12: Figures 3-A-d, 3-C-c and 3-F-c, it is not very clear how/whether the orange insets are related to the black and grey curves.

Addressing comment #11

The black, grey, and magenta curves are the “slow”, “fast”¹² and “fast”²¹ GLM coupling filters as defined by the GLMCC method (Kobayashi et al., 2019). These curves are indeed unrelated to the orange insets. The orange insets depict the PSP estimates based on the CoNNECT method (Endo et al., 2021).

Following this comment, we moved the magenta inset in **Fig. 3Ae** to the left, and added insets for the black, magenta, and grey curves.

References:

Endo, D. et al. A convolutional neural network for estimating synaptic connectivity from spike trains. Scientific Reports 11, 12087 (2021).

Kobayashi, R. et al. Reconstructing neuronal circuitry from parallel spike trains. Nature Communications 10, 4468 (2019).

Comment #12

Page 17: Figure 5-C-b, the y-axis label is incorrect, it should be corrected as eSTG.

Addressing comment #12

Following this comment, we fixed the y-axis label in **Fig. 5Cb** to read “eSTG”.

Comment #13

Page 22: line 536 & line 538, it would be better to replace all instances of S with S(t) to avoid confusion.

Addressing comment #13

Done (Methods, pg. 24, lines 565 and 567).

Comment #14

Page 22: line 544, please define the abbreviation ARP here, where it is first being used.

Addressing comment #14

Done (Results, pg. 10, line 211).

REVIEWERS' COMMENTS:

Reviewer #1 (Remarks to the Author):

The authors responded quite satisfactorily to my (minor) concerns. This is an important study.

Reviewer #2 (Remarks to the Author):

The issues raised by me were reasonably addressed. I have no further comments.

Reviewer #3 (Remarks to the Author):

Thank you for providing a very detailed rebuttal and for addressing all the issues in the original manuscript. The exposition of the theory and the applicability of the dcCCH method is significantly improved after the revisions. I have no further comments.